# Analyzing efficacy, stability, and safety of AAV-mediated optogenetic hearing restoration in mice

Burak Bali[1,2,3,*] ⓘ, Eva Gruber-Dujardin[4,*] ⓘ, Kathrin Kusch[1,5] ⓘ, Vladan Rankovic[1,3] ⓘ, Tobias Moser[1,6,7,8] ⓘ

AAV-mediated optogenetic neural stimulation has become a clinical approach for restoring function in sensory disorders and feasibility for hearing restoration has been indicated in rodents. Nonetheless, long-term stability and safety of AAV-mediated channelrhodopsin (ChR) expression in spiral ganglion neurons (SGNs) remained to be addressed. Here, we used longitudinal studies on mice subjected to early postnatal administration of AAV2/6 carrying fast gating ChR f-Chrimson under the control of the human synapsin promoter unilaterally to the cochlea. f-Chrimson expression in SGNs in both ears and the brain was probed in animals aged 1 mo to 2 yr. f-Chrimson was observed in SGNs at all ages indicating longevity of ChR-expression. SGN numbers in the AAV-injected cochleae declined with age faster than in controls. Investigations were extended to the brain in which viral transduction was observed across the organ at varying degrees irrespective of age without observing viral spread-related pathologies. No viral DNA or virus-related histopathological findings in visceral organs were encountered. In summary, our study demonstrates life-long (24 mo in mice) expression of f-Chrimson in SGNs upon single AAV-dosing of the cochlea.

## Introduction

Today, disabling hearing impairment affects 466 million people—roughly 5% of the world population—and estimates show that this number will double by 2050 (WHO, 2019). Sensorineural hearing impairment, acquired or congenital, due to disorders of the cochlea or the auditory nerve represents the most common form. Thus far there is no cure of sensorineural hearing impairment available. Current options for partial hearing restoration are hearing aids for moderate hearing impairment and hearing prostheses such as cochlear implants (CI) in case of profound hearing impairment and deafness. CIs directly stimulate the preserved auditory nerve, whereas auditory brainstem implants can be used in the absence of a functional auditory nerve. The CI with ~1 Mio users is considered the most successful neuroprosthesis. Nonetheless, the CI does typically not enable open speech comprehension in noisy environments. This is likely due to broad electric current spread from each contact electrode limiting the number of perceptually independent channels and, hence, sound frequency resolution (Jeschke & Moser, 2015; Dieter et al, 2020a).

Optogenetics has paved the way for spatiotemporally precise control of neuronal populations by sensitizing them to light using channelrhodopsins (Boyden et al, 2005). As a future option for improved hearing restoration by optical CIs (Weiss et al, 2016; Kleinlogel et al, 2020; Dieter et al, 2020a), optogenetics enabled optical control of SGNs that form the auditory nerve in various rodents (e.g., Hernandez et al [2014], Duarte et al [2018], Wrobel et al [2018], and Thompson et al [2020]). Optogenetic SGN stimulation showed a narrower spread of activation than electrical stimulation promising improved coding of spectral information in future oCIs (Hernandez et al, 2014; Dieter et al, 2019, 2020b; Keppeler et al, 2020). Moreover, behaviorally it enabled light-triggered auditory percepts in gerbils and rats (Wrobel et al, 2018; Jablonski et al, 2020 Preprint; Keppeler et al, 2020). Optogenetic hearing restoration requires the genetic modification of SGNs for expressing microbial opsins. For future clinical application, this genetic modification must be efficient and stable to provide users reliable optogenetic hearing for years after gene therapy. As in any drug development, the treatment should obey cellular specificity and lack harmful effects. Safety analysis should consider specific targeting of SGNs as well as unwanted immune response and neural degeneration in on- and off-target tissues. A previous study on optogenetic vision restoration investigated the stability and safety of adeno-associated virus (AAV)-mediated expression of channelrhodopsin 2 (ChR-2) in the rodent retina (Sugano et al, 2016). Physiological responses could be reliably elicited over 1 yr and histopathological analysis

[1]Institute for Auditory Neuroscience and InnerEarLab, University Medical Center Göttingen, Göttingen, Germany [2]Göttingen Graduate School for Neurosciences and Molecular Biosciences, University of Göttingen, Göttingen, Germany [3]Restorative Cochlear Genomics Group, Auditory Neuroscience and Optogenetics Laboratory, German Primate Center, Göttingen, Germany [4]Pathology Unit, German Primate Center, Göttingen, Germany [5]Functional Auditory Genomics, Auditory Neuroscience and Optogenetics Laboratory, German Primate Center, Göttingen, Germany [6]Auditory Neuroscience and Optogenetics Laboratory, German Primate Center, Göttingen, Germany [7]Auditory Neuroscience and Synaptic Nanophysiology Group, Max Planck Institute for Multidisciplinary Sciences, Göttingen, Germany [8]Cluster of Excellence "Multiscale Bioimaging: from Molecular Machines to Networks of Excitable Cells" (MBExC), University of Goettingen, Göttingen, Germany

Correspondence: tmoser@gwdg.de; vrankovic@dpz.eu
*Burak Bali and Eva Gruber-Dujardin contributed equally to this work.

showed little or no pathological changes of the retina that could be attributed to AAV administration or ChR-2 expression. However, despite local delivery AAV was systemically disseminated such that intestine, lung and heart tissue contained ChR-2 transcripts.

AAV-based gene delivery promises a stable and safe way to combat genetic diseases (Sahel & Roska, 2013; Keeler & Flotte, 2019; Kleinlogel et al, 2020). It provides non-pathogenic transport of gene-of-interest to target tissue where the genetic cargo remains episomal to the host genome. Despite being non-integrating, it can still achieve efficient long-term expression in postmitotic cells (Wang et al, 2019). Accordingly, AAVs are used in numerous gene therapy trials on the ear and the eye (Kleinlogel et al, 2020) and lately, three such gene therapies were approved by the European Medicines Agency and the US Food and Drug Administration. These are Glybera, LUXTURNA and ZOLGENSMA which are used for the treatment of genetic diseases lipoprotein lipase deficiency, inherited retinal disease, and spinal muscular atrophy, respectively (Keeler & Flotte, 2019; Nidetz et al, 2020). Gene therapies for cochlea, on the other hand, are emerging, too. A recently completed first trial (NCT02132130 [ClinicalTrials.gov, 2014]), aimed at hair cell regeneration by transdifferentiation of supporting cells via adenovirus-mediated misexpression of human Atonal transcription factor (*ATOH1*). Although a general improvement of auditory function was not achieved, importantly, no serious adverse effects were observed.

En route to clinical translation of optogenetic hearing restoration, we assessed efficiency, stability, and safety of AAV-based optogenetic manipulation of SGNs in mice. For this first study of stability and safety we turned to pressure injection of AAVs into the cochlea of neonatal mice, which is a popular delivery in preclinical studies of cochlear gene therapy. We choose AAV2/6 as a vector carrying the coding sequence for the red-light activated, fast gating f-Chrimson-eYFP (f-Chrimson) under the control of the neuron-specific human synapsin promoter. We opted for this approach based on our previous work that identified f-Chrimson to (1) enable near physiological response properties of SGNs, (2) to achieve good plasma membrane expression without additional trafficking signals, (3) to maintain functional SGN expression 9-mo after AAV injection, and (4) to reduce the risk of phototoxicity by red-light activation (Mager et al, 2018; Bali et al, 2021; Huet et al, 2021).

# Results

### f-Chrimson expression in SGNs upon a single cochlear AAV injection lasts for at least 2 yr

Using postnatal AAV pressure injection into the mouse cochlea we had previously tested for f-Chrimson expression in SGNs at young age (~1–4 mo old) and mid-age (9 mo) and demonstrated sustained expression that enabled robust optogenetic SGN stimulation (Mager et al, 2018). Here, we used immunohistochemistry at several time points over 24 mo for an extended longitudinal survey of f-Chrimson expression in mouse SGNs. Across all age-groups, apical and middle turns of the cochleae exhibited higher numbers of f-Chrimson–expressing SGNs compared with the basal turns in the injected cochleae (Fig 1A and B). On average, 1-mo-old mice had the

highest average density of f-Chrimson–expressing SGNs (~31 GFP$^+$ SGNs/ $10^4$ $\mu m^2$), twice as high as for injected cochlea analyzed at 3 mo or older (Table S1 and Fig 1). This decline likely reflects the loss of SGNs with age (see below and Sergeyenko et al [2013]). The counts of total (SGN) and f-Chrimson–expressing SGN of the youngest group indicates a transduction rate of 60% (Table S1) which is in line with previous estimates (Keppeler et al, 2018; Mager et al, 2018; Bali et al, 2021). Low levels of f-Chrimson expression were also observed in the contralateral non-injected cochlea: the average density of transduced SGNs was 1–2 cells in the sections of the apical, middle, basal turns of 1-mo-old mice. In conclusion, a large fraction of SGNs is transduced upon early postnatal AAV2/6 delivery and a sizable number of SGNs maintains f-Chrimson expression over the average mouse life span of 2 yr.

### Age-dependent decline in SGN density is accentuated in apical and middle turns

Age-related hearing loss is well established for C57Bl/6J mice that are known to start losing hearing by the age of 6 mo (Mikaelian et al, 1974; Kane et al, 2012). This age-related hearing loss initially manifests itself as degeneration of inner and outer hair cells at the high-frequency sound coding base of cochlea and a subsequent loss of basocochlear SGNs (Someya et al, 2009). Here, we aimed to disentangle SGN loss associated with age from potential neural loss caused by the optogenetic manipulation. Bilateral counting of immunofluorescent PV$^+$ somata per $10^4$ $\mu m^2$ (SGN density, Fig 2) in all age-groups of virus-injected animals and age-matched controls, either PBS-injected (3 mo old) or naïve (6 and 12 mo), revealed a general trend of SGN density decline with older age (Fig 2A). We first compared the SGN density between the injected and contralateral non-injected cochleae (Fig 2A–C) which did not differ from each other (two-way ANOVA with paired comparison, $P = 0.2623$). SGN densities averaged across all turns of the injected cochlea declined from 52.7 SGN bodies/$10^4$ $\mu m^2$ at 1 mo of age to 40, 46.8, 32.5, and 28.3 SGN bodies/$10^4$ $\mu m^2$ at 3, 6, 12, and 24 mo of age (one way ANOVA, $P = 0.0864$), respectively. However, this difference became statistically significant only at 24 mo of age (Dunnett's multiple comparison for 1 versus 24 mo, $P = 0.0479$). Contralateral non-injected cochleae had similar average densities of 49.1, 45.2, 43.6, 45.2, and 25.8 SGN bodies/$10^4$ $\mu m^2$ for 1, 3, 6, 12, and 24-mo-old age-groups, respectively, and a similar but significant decline with age (one-way ANOVA, $P < 0.0001$ and Dunnett's multiple comparison for 1 versus 24 mo, $P < 0.0001$). Given the minute transduction rate in non-injected ears (Fig 1), this argues against a major contribution of optogenetic manipulation to the SGN loss in the non-injected cochleae.

Mean SGN densities in older age-groups (12 and 24 mo) tended to be lower for apical and middle cochlear turns of AAV-injected cochleae compared with non-injected cochleae. Considerable intra-group variation together with the small sample size might have kept us from documenting statistical significance for the apical turn, but was detected for the middle cochlea turn (Fig 2C, two-way ANOVA with paired comparison for apical turn: $P = 0.0883$ and for middle $P = 0.0336$). However, pairwise comparisons of average SGN densities in the injected cochlea between 1- versus 12-mo-old and 1- versus 24-mo-old showed a significant decrease (Tukey's multiple comparisons test, apical 1 versus 12 mo $P = 0.0011$, apical 1 versus 24 mo $P = 0.02$; middle 1 versus 12 mo $P = 0.0165$,

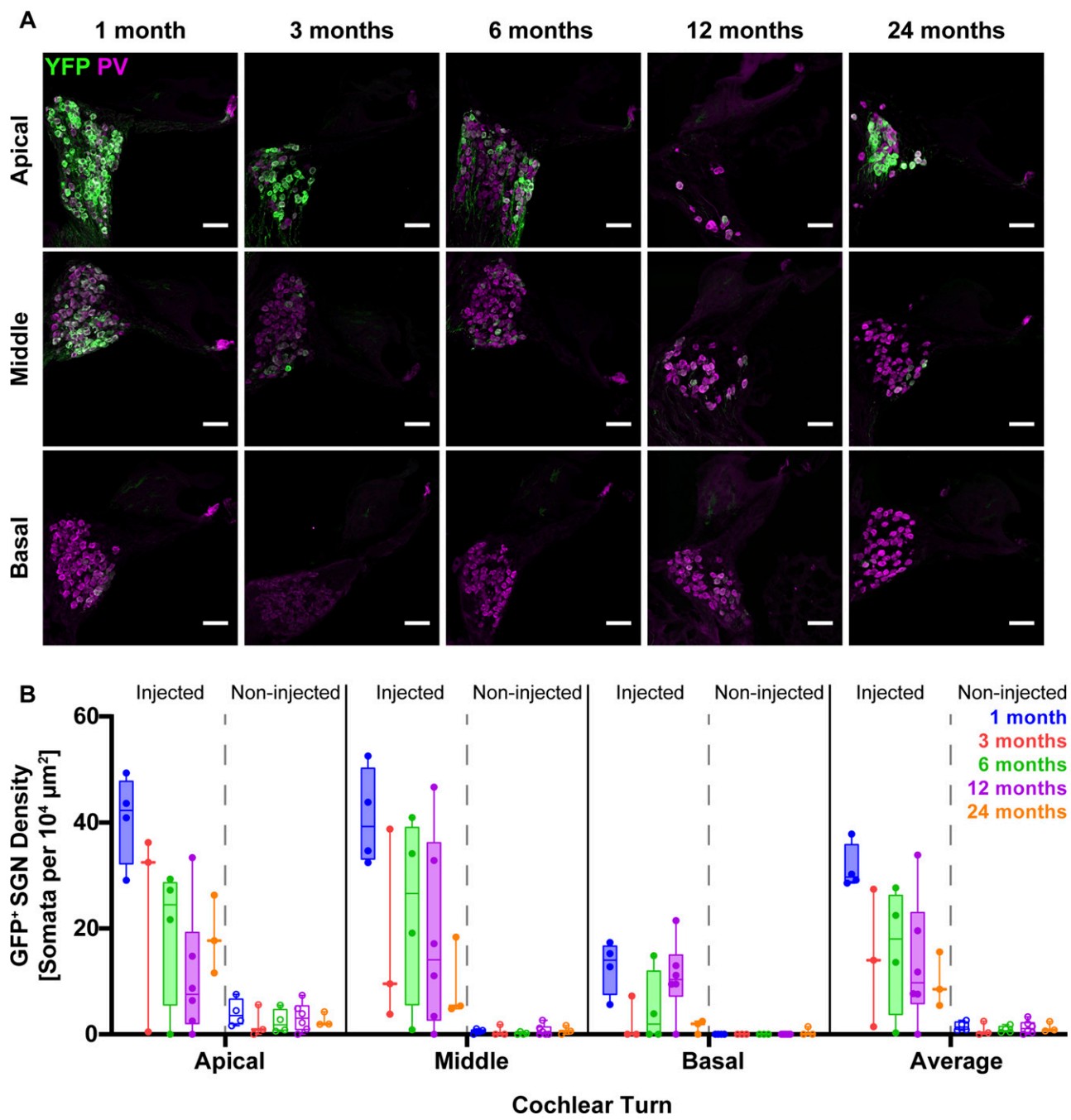

**Figure 1. Expression of f-Chrimson in spiral ganglion neurons of the mouse cochlea.**
**(A)** Exemplary confocal images of immunofluorescently labeled mid-modiolar cryosections (PV as context marker, GFP for detecting f-Chrimson-eYFP) from postnatally AAV-injected cochleae for the different age-groups. Scale bar: 50 $\mu$m. **(B)** Density of GFP$^+$ spiral ganglion neuron averaged across all cochlear turns for the injected (filled boxes) and contralateral non-injected (open boxes) cochleae of all age-groups. Box–whisker plots with upper and lower limits of the boxes representing 75% and 25% confidence intervals. Horizontal lines in the boxes indicate the median of the corresponding dataset. $N_{1-mo} = 4$, $N_{3-mo} = 3$, $N_{6-mo} = 4$, $N_{12-mo} = 6$, $N_{24-mo} = 3$ mice.

middle 1 versus 24 mo $P = 0.1591$), whereas in the contralateral non-injected ear no significant difference in SGN density was observed.

The average SGN density of 40 SGN bodies/10$^4$ $\mu$m$^2$ in AAV-injected cochleae from 3 mo old mice (n = 3) compared with 47.3 SGN bodies/10$^4$ $\mu$m$^2$ the PBS-injected control (n = 2 mice). The average SGN densities of AAV-injected cochleae of 6- and 12-mo-

old animals (46.8 and 32.5 SGN bodies/10$^4$ $\mu$m$^2$, n = 4 mice and n = 6 mice, respectively) seemed comparable with average SGN densities in age-matched naïve cochleae (43.8 and 39.5 SGN bodies/10$^4$ $\mu$m$^2$, respectively, n = 2 mice each).

In addition to SGN density estimates, comparative semi-quantitative histopathological scoring of HE stained cochlea

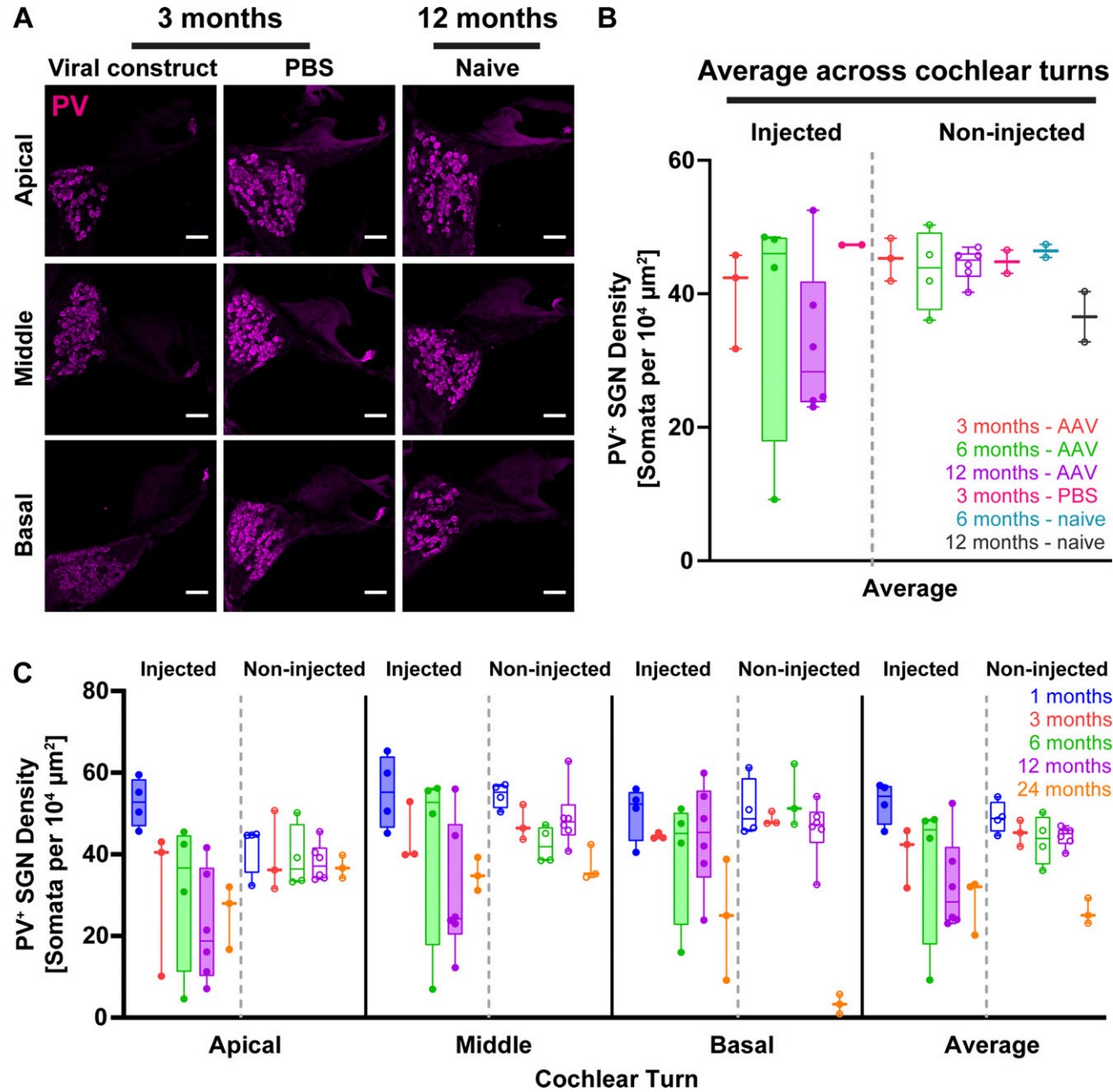

**Figure 2.  Spiral ganglion neuron (SGN) density of f-Chrimson–injected cochleae in comparison with negative controls.**
**(A)** Confocal images of mid-modiolar cross-sections from virus-injected, PBS-injected, and naïve cochleae from 3- to 12-mo-old mice. Scale bar: 50 $\mu$m. **(B)** SGN densities estimated as number of parvalbumin-immunofluorescence positive cells, PV+, per area of Rosenthal's canal (see the Materials and Methods section) of modiolar turns from the injected cochleae of AAV-f-Chrimson–, PBS-injected, and naïve animals for the age-groups of 3-, 6-, and 12-mo-old mice. **(C)** Border designation of the plots is as in (C). $N_{3-mo}$ = 3, $N_{6-mo}$ = 4, $N_{12-mo}$ = 6 mice for AAV-f-Chrimson; $N_{3-mo}$ = 2 mice for PBS; $N_{6-mo}$ = 2 mice and $N_{12-mo}$ = 2 mice for naïve group. **(C)** SGN densities across all cochlear turns from the injected (filled boxes) and contralateral non-injected (open boxes) cochleae across the age-groups are plotted in box–whisker graphs. Upper and lower limits of the boxes represent 75% and 25% of the dataset, respectively. Horizontal lines in the boxes indicate the median, whereas the whiskers stand for maximum and minimum of the corresponding dataset. $N_{1-mo}$ = 4, $N_{3-mo}$ = 3, $N_{6-mo}$ = 4, $N_{12-mo}$ = 6, $N_{24-mo}$ = 3 mice.

sections from different age-groups (1, 3, 6, 12, and 24 mo) showed similar trends of age-related changes. Fig 3 shows exemplary micrographs comparing the virus-injected (Fig 3A) and the contralateral, non-injected cochlea (Fig 3B) of a 12-mo-old mouse. Apical and middle turns of AAV-injected cochleae from 12- to 24-mo-old mice revealed lower scores for neuron density and increased

interstitial vacuolation compared with the contralateral, non-injected side, as well as to PBS-injected and naïve cochleae (Fig S1). However, deposition of cellular/karyorrhectic debris as a morphological indicator of single cell necrosis/apoptosis differed only slightly between apical and middle parts of virus-injected and contralateral cochleae, and not at all compared with age-matched

controls. Basal turns of injected and control cochleae did not show obvious differences in SGN density, interstitial vacuolation, and deposition of cell debris across all age-groups. A panel with micrographs of HE stained cochleae (injected versus non-injected side) from all mice analyzed is provided in the supplementary material (Table S3).

## f-Chrimson expression in the brain following postnatal cochlear AAV-injection

For investigating potential spread of AAV-mediated f-Chrimson expression to the nervous system following postnatal intra-cochlear injection, IHC-stained, paraffin-embedded, coronal brain sections from different age-groups were analyzed along the rostro-caudal axis (Fig 4A). Block I (Front) comprised cortical und subcortical forebrain areas, block II (Mid) covered mainly mid-cortical and diencephalic regions, such as visual and auditory cortices, hippocampus, and thalamus. Cerebellum, inferior colliculi, and auditory brainstem were covered by block III (Hind). GFP[+] immunolabeling, indicative for f-Chrimson expression, could be observed throughout the entire brain, as shown in Table 1. GFP[+] perikarya and projections were present bilaterally from subcortical areas, such as the auditory brainstem and cochlear nucleus, up to cortical regions, such as auditory and retrosplenial cortices (Fig 4B). In brain sections of all naïve control animals, no specific DAB signal was present.

## f-Chrimson expression is not associated with histopathological changes in the brain

Histopathological evaluation of serial HE stained brain sections from virus-injected mice at the five different ages analyzed (1–24

mo old) as well as from PBS-injected (3-mo old) and naïve control animals (6 and 12 mo old) focused on semiquantitative scoring for neuronal degeneration, perivascular inflammatory cells infiltrates ("perivascular cuffing"), and gliosis. Brain regions were compared ipsi- and contralateral to the side of the AAV-injected cochlea. No consistent differences could be detected between virus-injected and sham- or non-injected control groups, respectively (Fig S2). However, we observed a mild bilateral increase in perivascular cuffing and gliosis in blocks II and III of the oldest animals (24 mo).

## Age-related increase in cerebral T-cell infiltration and activated microglia is independent of cochlear AAV-injection and cerebral f-Chrimson expression

For detection of potential neuroimmunological or inflammatory changes associated with viral transduction or expression of the microbial opsin f-Chrimson, consecutive coronal brain sections from all three brain blocks were IHC-stained for CD3 as a mature pan-T-cell marker, and IBA-1 for microglia labeling. The brain is almost devoid of CD3-positive T cells (<1 per mm$^2$) in healthy mice younger than 12 mo old (Ferretti et al, 2016) but 2–3 cells per mm$^2$ can be found in healthy old mice (24 mo). In models of neuro-degenerative disease, such as Alzheimer's, infiltration of T cells rises 3–4 folds compared with age-matched mouse group (Ferretti et al, 2016). Therefore, CD3 is a suitable marker for detection of potential age- and f-Chrimson–related neuroimmunology. In our study, CD3$^+$ T cells were also scarce in younger age-groups (<12 mo), whereas in the oldest virus-injected group (24 mo, virus), they were frequently encountered (Fig 5A). However, GFP$^+$ brain regions did not exhibit increased numbers of CD3$^+$ cells when compared with GFP$^-$ areas (Fig 5B). Further quantification showed that 1-, 6-, and

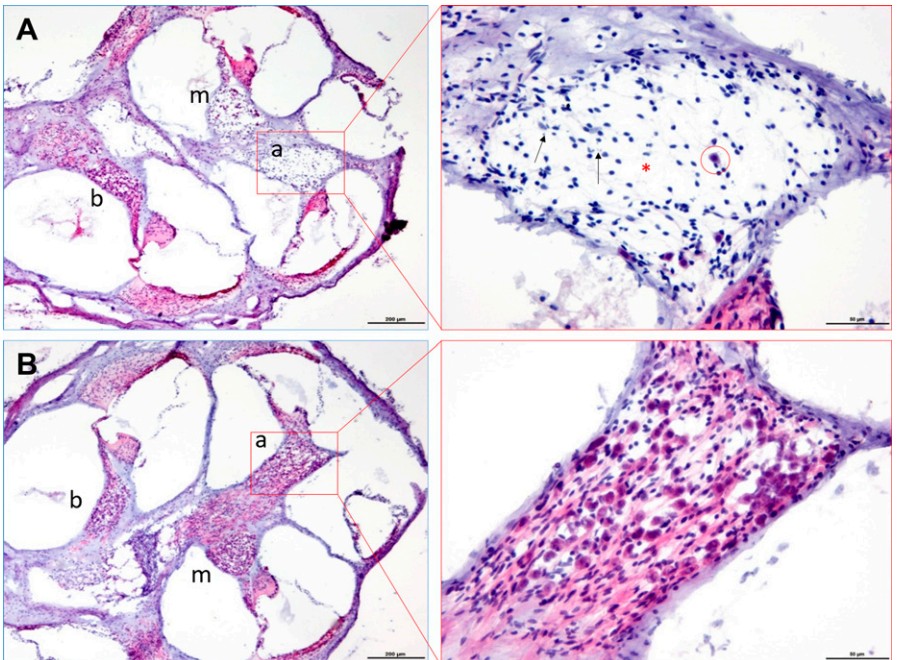

**Figure 3.  Histopathological analysis of the cochlea.**
**(A, B)** Light micrographs of a HE stained cochleae of a 12-mo-old mouse (#652279): AAV-injected (A) side showing an extreme example of spiral ganglion neuron loss (circle represents one of the remaining spiral ganglion neurons), accompanied by interstitial vacuolation (asterisk) and some cellular debris (arrows) in the apical (a) and middle (m) modiolar turns, compared with the contralateral, non-injected side (B). Histological appearance of basal (b) parts was more similar on both sides; scale bars left: 200 μm; right: 50 μm.

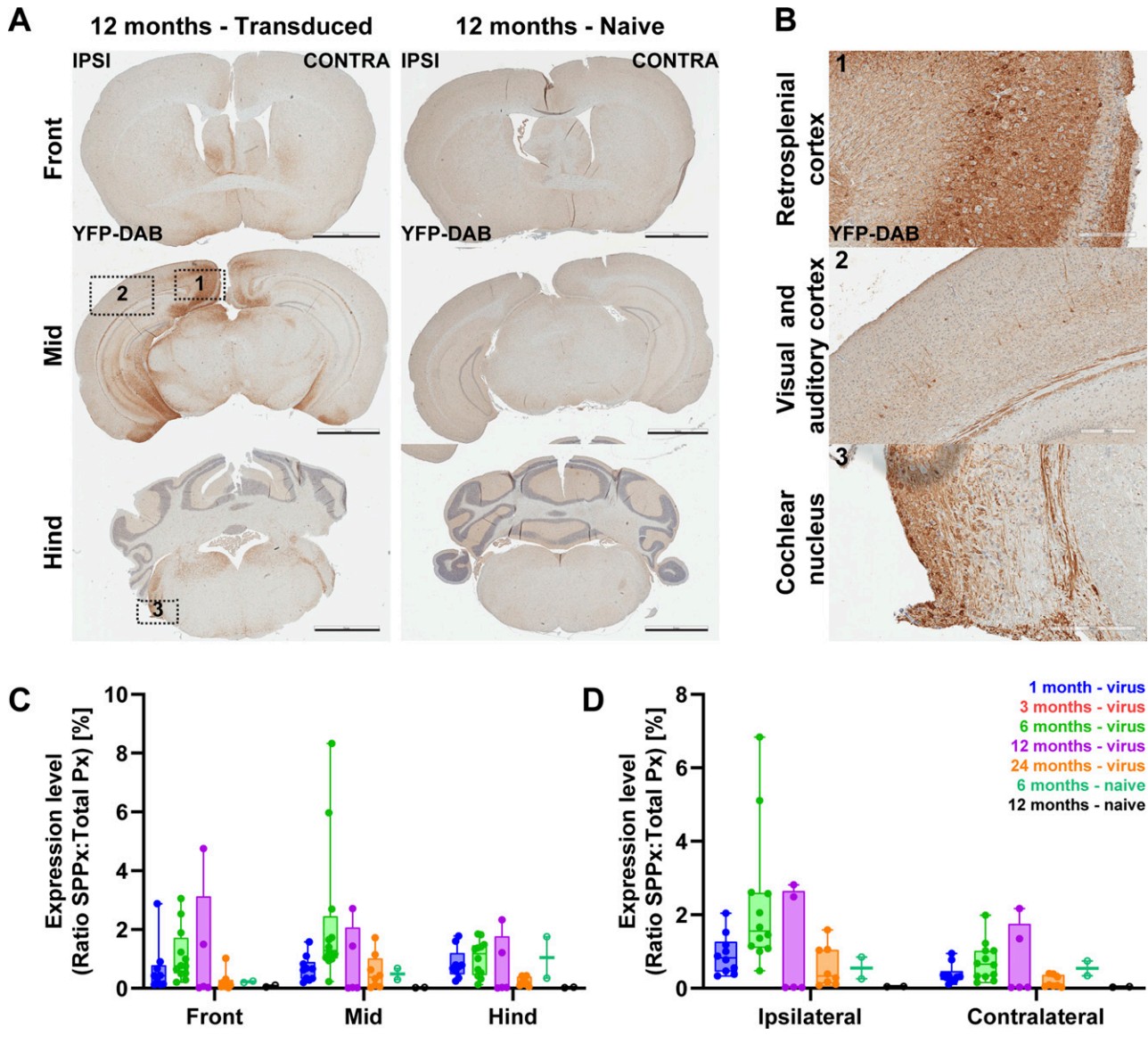

**Figure 4. Off-target expression of viral f-Chrimson construct.**
**(A)** Exemplary light microscopy images of coronal sections of transduced and naïve mice at the age of 12 mo are shown. Chromogenic DAB (brown) labels the regions where f-Chrimson expression is present on the three brain blocks along rostro-caudal axis. Scale bars: 2 mm. **(A, B)** Zoom-in images of the numbered regions from (A). Scale bars: 200 $\mu$m. **(C, D)** Quantification of expression of f-Chrimson along rostro-ventral (C) or lateral (D) axes.

12-mo-old animals of the AAV-injected group had ≤1 cell per mm[2], whereas T-cell numbers were significantly higher along the rostro-caudal axis of 24-mo-old mice (Fig 5C, ordinary two-way ANOVA, $P$ = 0.0013; N = 3 mice per group). Age-matched naïve controls of 6- and 12-mo-old mice had comparably low densities of CD3[+] cells (ordinary two-way ANOVA, $P$ = 0.5015; N = 2 mice per group) supporting the observed increase in T-cell infiltration being rather age-related than associated with intracochlear AAV injection. Correspondingly, T-cell counts of 3-mo-old PBS-injected mice did not differ from those of 1-mo-old virus-injected mice (ordinary two-way ANOVA, $P$ = 0.2116, Fig 5C) and bilateral comparison of T-cell densities also revealed no obvious differences between ipsi- and contralateral hemispheres (Fig 5D).

In a healthy brain, microglial cells reside in a finely ramified state (arachnoid morphology). Microglial activation occurs in the presence of pathogens or degenerative changes in neural tissue. Activated microglia undergo distinct morphological changes by shortening and thickening of cellular processes towards a rather amoeboid shape (Felsky et al, 2019). Microglia density and activation was evaluated for detection of potential inflammatory responses to AAV-mediated f-Chrimson expression, respectively. For this purpose, density, distribution and staining intensity of IBA-1[+] cells was determined in digital IHC stained coronal brain sections from AAV-injected and naïve mice of different age (Fig 6). Compared with highly inflamed, positive control tissue (Fig 6A), microglia density was much lower in brain sections of AAV-transduced and

**Table 1.** Brain regions with regularly observed f-Chrimson expression.

| | Brain regions/structures |
|---|---|
| Block I (front) | (Anterior) cingulate, (pre)limbic, orbital, primary motor and somatosensory, olfactory-related such as taenia tecta, and olfactory tubercle |
| Block II (mid) | Retrosplenial (layer 5 and/or 6), hippocampus especially dentate gyrus, visual and auditory areas, piriform, cortical-amygdalar areas, thalamus, and hypothalamus |
| Block III (hind) | Cerebellum, inferior colliculus, cochlear nucleus, trapezoid body, olivary complex, ascending fibers to vestibular nucleus |

Quantitative analysis of f-Chrimson expression in the brain across all age-groups revealed an overall low DAB signal (<3% of all pixels), both along rostro-caudal and medio–lateral axes (Fig 4C and D). Comparison between both hemispheres showed slight emphasis on the ipsilateral side of virus-injection, especially in 6-mo-old mice.

naïve mice with a well-preserved arachnoid morphology (Fig 6Bi and Bii). However, some degree of age-related microglia activation was present, both in AAV-injected and control groups. For quantification, we defined the coverage of microglial activation as the total number of strongly positive brown pixels divided by the area of interest. As with T-cell infiltration, an age-related increase in microglia activation was detected in transduced mice (Fig 6C), which only reached significance when comparing 1- versus 12-mo-old and 1 versus 24-mo-old age-groups (Tukey's multiple comparison test, e.g., for mid brain $P_{1vs12} = 0.0014$, $P_{1vs24} < 0.0001$). However, microglia activation observed in 24-mo mice was still much lower than in the inflamed positive control sample. Overall microglia activation was comparable between samples of age-matched transduced mice and naïve controls (Tukey's multiple comparison test, 6 mo old and mid brain, $P = 0.9863$: 12 mo old and mid brain, $P = 0.9999$) and between brain hemispheres ipsi- and contralateral to the injected ear (Fig 6D; ordinary two-way ANOVA, $P = 0.5111$).

### f-Chrimson DNA was not detected in inner organs of mice with postnatal cochlear AAV2/6 injection

Given the spread of AAV beyond the postnatally injected ear with f-Chrimson expression in the contralateral ear and brain, potential biodistribution of AAV to internal organs was surveyed by conventional PCR on native heart, lung, liver, kidney and spleen tissue. A 917 base pair-long DNA strip that partially covered coding sequences of f-Chrimson and eYFP, was targeted and amplified (Fig 7A). The viral f-Chrimson construct was not detected in any of the inner organs tested (Fig 7B, lanes 6–10) but in all f-Chrimson injected cochleae. Only in one 1-mo-old animal, PCR yielded bands for both cochleae (Fig 7B, lanes 4–5). We used template plasmid DNA (lane 1) as a positive control that showed the expected size band of ~900 bp, yet there also was a smear formation. As the positive control for DNA isolation (lane 3), primers for RIM-binding protein-2 genomic DNA was used which amplified a ~350 bp band. Last, the negative control (lane 2) for PCR, where distilled water was used as the template interestingly, also sometimes produced a faint band around 1,100 bp which is above the expected size of 917 bp.

### AAV-mediated f-Chrimson delivery is not associated with histopathological changes in internal organs

Several organs (lung, heart, liver, spleen, and kidney) of virus-injected mice from three age-groups (6, 12, and 24 mo) as well as from pbs-injected and naïve control groups were sampled for microscopic evaluation of the general health status. Results of semiquantitative scoring (see the Materials and Methods section) for organ-specific histopathological changes, such as mononuclear and granulocytic inflammatory cell infiltrates, cellular degeneration, interstitial fibrosis, ectasia of renal tubules, or lymphoid hyperplasia, activation, and splenic extramedullary hematopoiesis (EMH) are provided in Table 2. Besides some age-related increase in EMH, and mildly elevated inflammatory/degenerative changes in liver and kidney tissues, none to mild unspecific findings were similarly distributed over all virus-injected and control groups investigated.

## Discussion

This work investigated efficacy, stability, specificity, and safety of AAV-mediated optogenetic manipulation of SGNs in mice. We used a single unilateral injection of AAV2/6 carrying f-Chrimson-eYFP under the control of the human synapsin promoter into the early postnatal cochlea and analyzed both cochleae, brain, and inner organs in a longitudinal study comprising the entire mouse life span.

### Virus-mediated optogenetic manipulation of SGNs

In this study, expression of f-Chrimson was maintained in a good fraction of SGNs (>30%) for up to 24 mo—the average life span of a laboratory mouse (Flurkey & Currer, 2004). First clinical results on the use of optogenetics for vision restoration (NCT02556736 and NCT03326336) indicate favorable safety profiles of AAV-mediated optogenetic manipulation of retinal ganglion cells and document promising preliminary outcome results on a patient with retinitis pigmentosa (Sahel et al, 2021). Yet, these are early days of translating optogenetics to clinical applications and each of them requires careful analysis of safety, efficacy, and stability in preclinical and, where feasible, in clinical studies. We chose mouse as a rodent model for studying hearing, its impairment and restoration (Petit et al, 2001; Steel & Kros, 2001; Moser & Starr, 2016; Kleinlogel et al, 2020). We chose C57Bl/6J mice as a model for two reasons. First, C57Bl/6J mice are typically used in mouse mutagenesis. Consequently, most of the available preclinical mouse studies on genetic hearing loss, gene therapy, and optogenetic hearing restoration have used this strain such that the results obtained in the

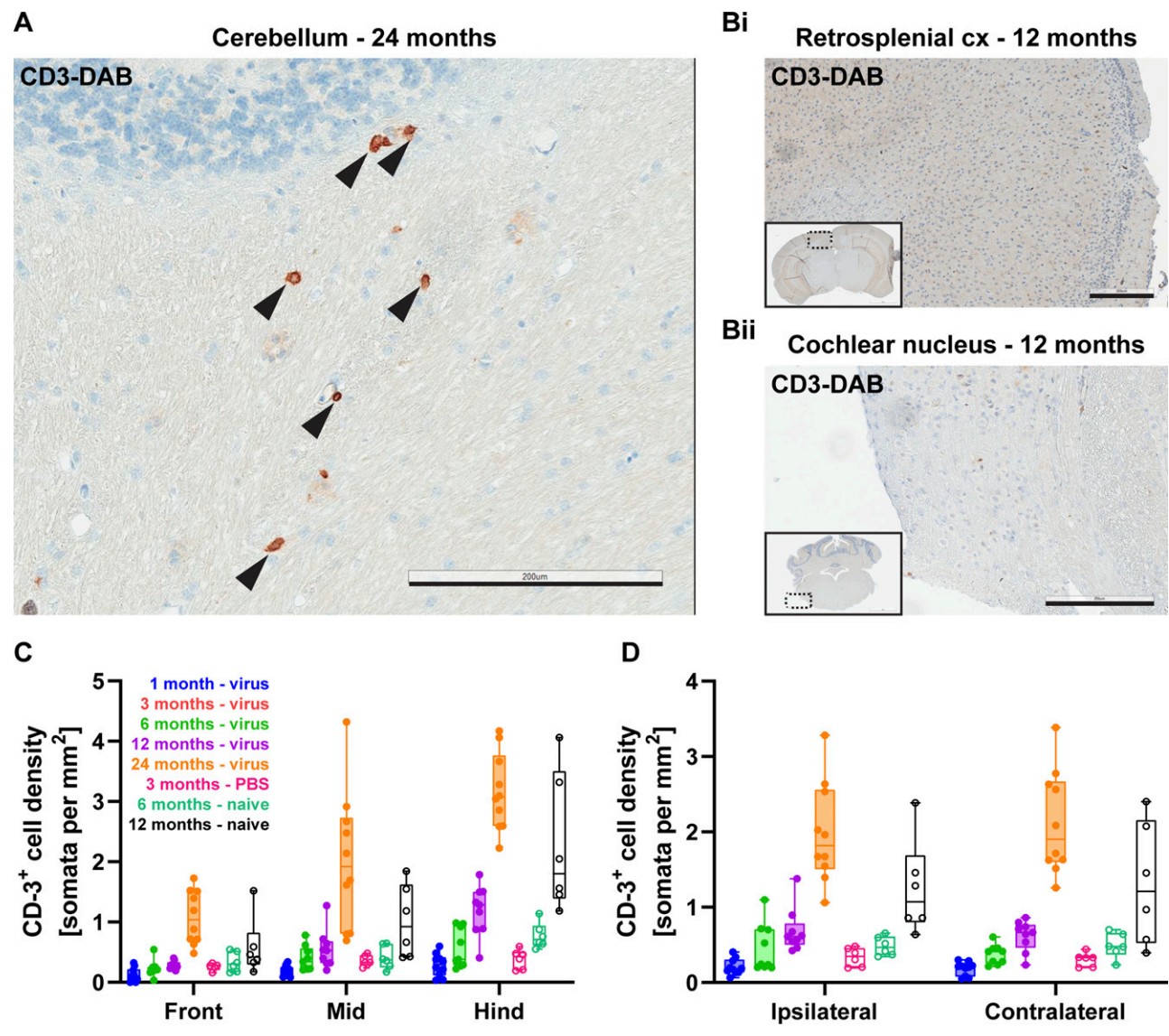

**Figure 5. Age-related T-cell infiltration in the brain.**
**(A)** Exemplary CD3⁺ T-lymphocytes (arrowheads) are shown in the white matter of cerebellum from a 24-mo-old mouse. Scale bar: 200 $\mu$m. **(B)** CD3 histology on retrosplenial cortex (Bi) and cochlear nucleus (Bii) corresponding to f-Chrimson⁺ regions from Fig 4B. Insets show the relative location of the zoom-in image to the whole coronal section. Scale bars: 200 $\mu$m. **(C, D)** Average T-cell densities are plotted either along rostro-caudal (C) or lateral (D) axes.

present study on C57Bl/6J can be related. Second, C57Bl/6J mice serve as a model of age-related hearing loss due to a mutation in the cadherin-23 gene: are known to start losing hearing by the age of 6 mo (Mikaelian et al, 1974; Kane et al, 2012). Hence, they represent a valuable disease model and our prior work showed that optogenetic stimulation of the auditory nerve was as efficient at the age of 9 mo as at 2–3 mo, whereas acoustic hearing was strongly degraded in the older C57Bl/6J mice (Mager et al, 2018). This age-related hearing loss initially manifests itself as degeneration of inner and outer hair cells at the high-frequency sound coding base of cochlea and a subsequent loss, especially of basocochlear SGNs (Someya et al, 2009). We consider the use of a model of age-related hearing loss valuable for testing stability and safety of AAV-mediated optogenetic

manipulation of the SGNs, but stress that disentangling genetic mechanisms of SGN death from potential harm related to AAV-dosing is far from trivial.

Here, we embarked on the highly efficient early postnatal AAV-mediated cochlear gene delivery (e.g., Akil et al [2012], Askew et al [2015], Jung et al [2015], Landegger et al [2017], Keppeler et al [2018], Al-Moyed et al [2019], and Rankovic et al [2021]) primarily to evaluate the stability of optogenetic SGN manipulation. This is highly relevant given the non-integrative nature of the AAVs and the possibility that the transcription from the episomal viral DNA could run-down over time. We chose to express the red-light activated f-Chrimson that serves as a candidate for future clinical optogenetic hearing restoration. f-Chrimson offers (i) rapid closing

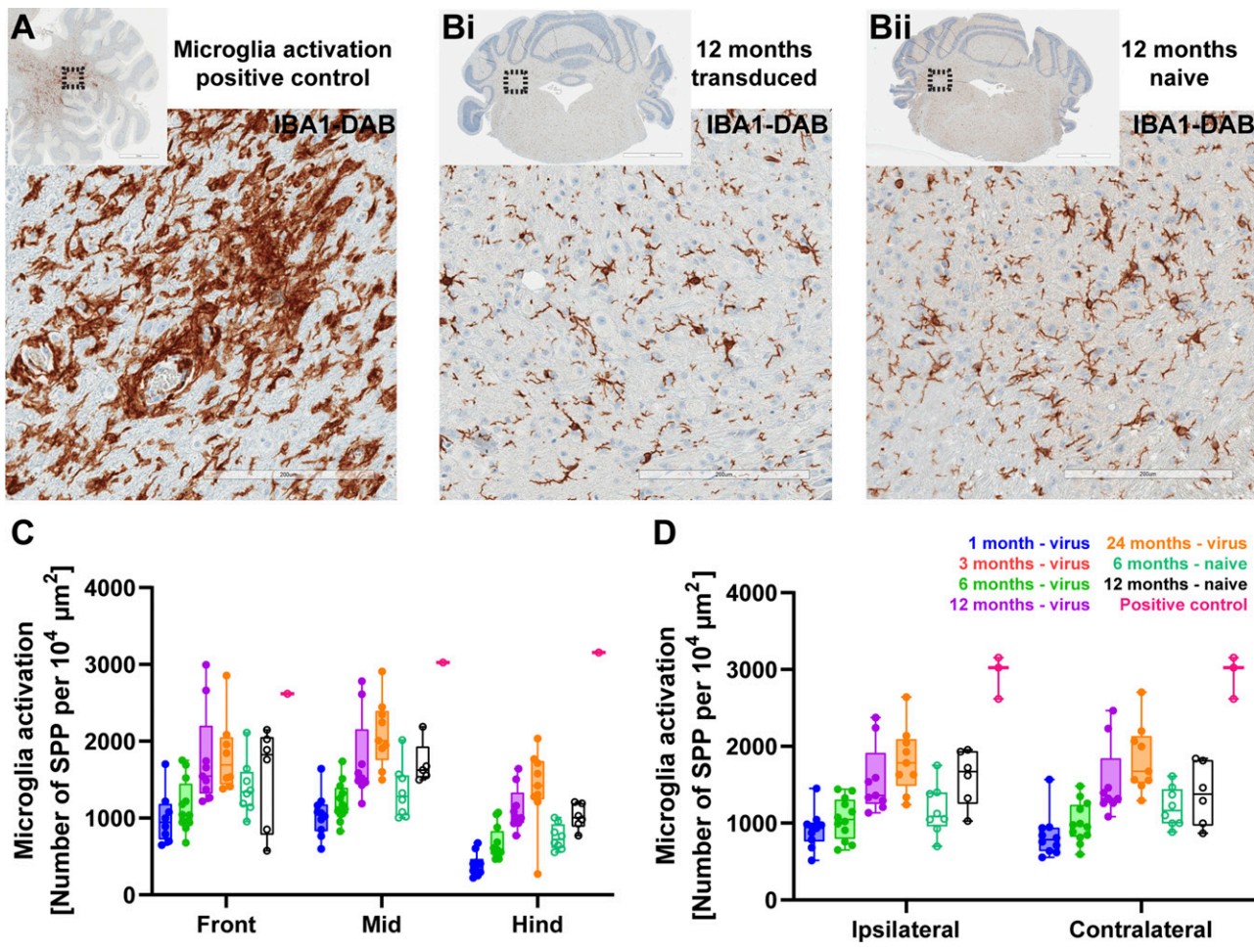

**Figure 6. Microglia activation profile.**
**(A, B)** Exemplary light microscopy images of IBA-1 labeled cerebellar tissue from helminth-infected (A), transduced (Bi), and naïve (Bii) animals. Insets show the overview images. Scale bars: 200 $\mu m$, and 2 mm for insets. **(C, D)** Microglia activation, number of strongly positive pixels per $10^4$ $\mu m^2$, plotted for rostro-caudal (C) and lateral profiles (D) for virus-injected, naïve, and helminth-infected neural tissue as positive control.

kinetics for stimulation of SGNs with good temporal fidelity, as well as (ii) excellent photocurrents due to robust plasma membrane expression and (iii) lowers the risk of phototoxicity (Mager et al, 2018; Bali et al, 2021; Huet et al, 2021). Likewise, AAV2/6 and the human synapsin promoter are of interest for clinical translation given potent SGN transduction with good specificity at least within the cochlea (Mager et al, 2018; Wrobel et al, 2018). To boost expression and aid the analysis, we used the woodchuck hepatitis virus posttranscriptional regulatory element (WPRE) and the bovine growth hormone (bGH) polyadenylation sequences as well as the eYFP tag fused to the C terminus of f-Chrimson (Gauvain et al, 2020). Whereas our analysis was confounded by the age-related, most likely degenerative loss of SGNs, it indicates a favorable stability of transcription, translation, and ensuing plasma membrane expression upon a single administration of AAV to the cochlea. Moreover, within the cochlea, the expression was limited to the SGNs, the only neural population with cochlear cell bodies and no expression was found in hair cells or glial/supporting cells, which we attribute to the choice of serotype and promoter. We did not

differentiate type I and II SGNs or type I subtypes (Petitpré et al, 2018; Shrestha et al, 2018; Sun et al, 2018; Li et al, 2020), which might be a topic for future studies. Specifically, expressing different types or amounts channelrhodopsins to type I SGN subtypes might offer an interesting perspective to increase the output dynamic range of optogenetic sound encoding (Bali et al, 2021).

In this study, we also aimed to get a first glimpse into the long-term safety of AAV-mediated optogenetic SGN manipulation. For this purpose, we studied the neural status of the AAV-injected cochlea in comparison to the contralateral, non-injected cochlea of the same animals and to age-matched control cochleae of PBS-injected and naïve mice from different age-groups. The postnatal pressure injection did not cause any obvious immediate SGN loss that we would have expected to detect at the earliest time point checked (1 mo). Besides that, analysis of SGN densities showed an age-dependent SGN loss which, for the injected ear, was significantly greater when compared with the non-injected contralateral ear at the age of 12 mo. However, this estimated SGN density of the injected ears at 12 mo was statistically indistinguishable from that

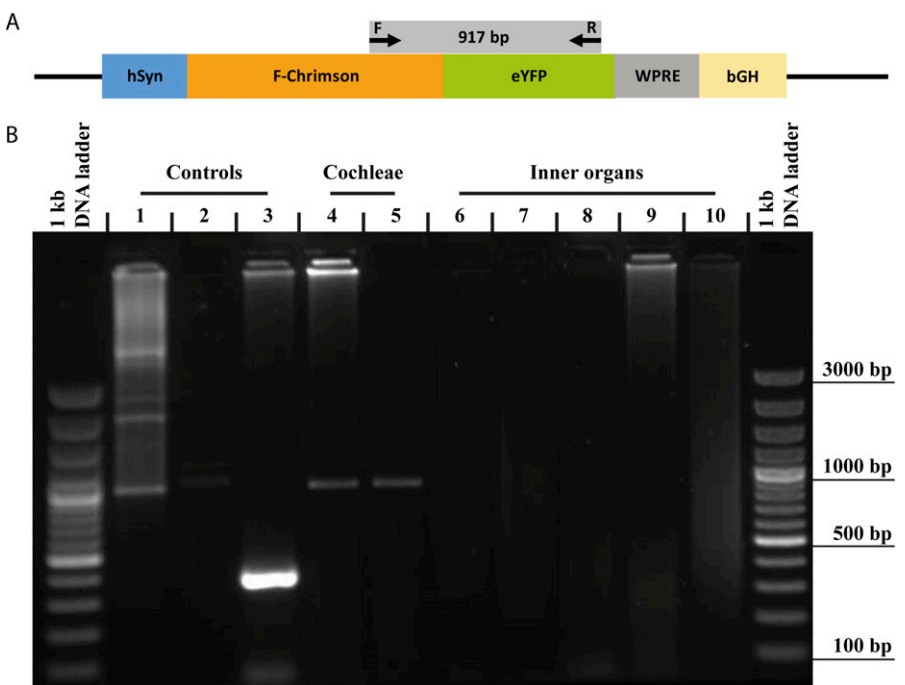

A

**Figure 7. f-Chrimson DNA was not detected in inner organs of mice.**
**(A)** pAAV vector used in the study. Arrows indicate forward (F) and reverse (R) primers. **(B)** Agarose gel image of PCR products amplified from template plasmid DNA (pAAV-hSyn-f-Chrimson-eYFP-WPRE-bGH) as positive control for amplified site (1), double-distilled H$_2$O as PCR negative control (2), tissue DNA sample extracted from contralateral non-injected cochlea as DNA isolation control (3), tissue DNA samples (dashed rectangle) extracted from contralateral non-injected cochlea (4), injected cochlea (5), heart (6), lung (7), liver (8), kidney (9), and spleen (10) are shown. All tissue samples are from a 1-mo-old transduced mouse. Expected PCR product size for cochleae and the inner organs is 917 bp (4–10) whereas it is 367 bp for DNA isolation control (3).

of cochleae from non-injected or PBS-injected mice. Given the greater statistical power of the intra-individual comparison and with respect to the more pronounced histopathological changes of the spiral ganglion in the injected ear, a potential negative, substance-related long-term effect of the AAV-mediated opto-genetic manipulation cannot be ruled out and needs further investigation. The age-dependent decline of SGN density seemed to be pronounced in apical and medial turns of the AAV-injected cochleae from elderly mice of 12 and 24 mo, which together with the apicobasal gradient could indicate an adverse effect of the AAV-treatment. Future studies in animals lacking basoapically progressing cochlear pathology found in C57BL6J mice will help to further evaluate adverse effects of the AAV-treatment. Such effects could imply an additional metabolic demand and/or proteostatic stress for those neurons continuously expressing the opsin and/or the reporter protein YFP, which could contribute to accelerated cellular degeneration and necrosis/apoptosis over time. Other possible reasons for SGN loss would be primary neural degeneration due to the experimental procedure (pressure injection), direct viral toxicity by the injected AAV, as previously described (Hirsch et al, 2011; Johnston et al, 2021; Suriano et al, 2021 *Preprint*) or neuroimmunological attack of transduced SGNs. In our study, the observed accelerated SGN loss in apical and middle turns of the ganglion became most evident in the older age-groups and was not conspicuous in one and 3 mo old mice. Hence, a primary, injection-related cause or directly AAV-induced apoptosis seem rather unlikely because this might have led to an earlier decline in SGN density. However, for disentangling AAV- or transgene-related effects, future studies could consider additional control groups that received AAVs expressing YFP alone, AAVs expressing no transgenes or empty AAV capsids. With our current analysis, we could not rule out a possible neuroimmunological pathogenesis of accelerated

SGN loss in the apical and middle turns as histopathological scoring solely relied on standard HE-stained cochlear sections. However, we consider this rather unlikely because no obvious histopathological signs of inflammation or noticeable leucocyte infiltration could be observed across all age-groups. Still, to reliably address possible inflammatory cell infiltrates, additional immunostaining of the spiral ganglia with different immune cell markers and their quantification would be necessary, which is planned for our future studies also including non-human primates.

**Spread of virus from the cochlea upon pressure injection in the first postnatal week**

(Sergeyenko et al, 2013) f-Chrimson expression in the injected cochlea was specific to SGNs suggesting that the choice of capsid and promoter has potential for testing clinical translation if administration can limit the AAV to the injected cochlea. A significant, although lower f-Chrimson, expression was also found in SGNs of the contralateral non-injected ear and in the brain. Hence, we postulate viral spread from the postnatally injected ear to the cerebrospinal fluid space via a competent cochlear aqueduct as previously described (Lalwani et al, 1996). This view is further supported by a recent study of injecting dye through the round window membrane into the cochlea in neonatal and adult mice showed a spread of the dye to brain via the cochlear aqueduct (Talaei et al, 2019). Yet, we did not detect obvious histopathology owing to the AAV-mediated optogenetic manipulation in the brain. To study spread beyond the nervous system we additionally used PCR-based analysis of inner organs which did not detect the viral f-Chrimson construct. This is consistent with a previous inner ear gene therapy study (Kho et al, 2000) that neither detected viral spread to the inner organs. Although negative

**Table 2. Average scores of inflammatory and degenerative lesions in several organs from different virus-injected (-v) age-groups, sham-injected (-pbs), and naïve (-n) control groups.**

| | | AAV | | | Control | | |
|---|---|---|---|---|---|---|---|
| | | 6 mo-v | 12 mo-v | 24 mo-v | 3 mo-pbs | 6 mo-n | 12 mo-n |
| Lung | Mononuclear leucocytes | (+) | (+) | (+) | (+) | (+) | (+) |
| | Granulocytic leucocytes | (+) | (+) | (+) | (+) | (+) | (+) |
| | Alveolar edema | (+) | — | — | (+) | (+) | — |
| Heart | Mononuclear leucocytes | (+) | — | (+) | — | — | (+) |
| | Fibrosis | — | — | — | — | — | — |
| Liver | Mononuclear leucocytes | (+) | (+) | + | (+) | (+) | + |
| | Granulocytic leucocytes | (+) | — | (+) | — | — | (+) |
| | Hepatic degeneration | (+) | + | + | (+) | (+) | (+) |
| | Interstitial fibrosis | — | (+) | (+) | — | — | — |
| Kidney | Mononuclear leucocytes | (+) | + | + | (+) | (+) | — |
| | Granulocytic leucocytes | — | (+) | (+) | — | — | — |
| | Tubular degeneration | (+) | — | (+) | — | — | — |
| Spleen | PALS hyperplasia | ++ | (+) | ++ | + | (+) | ++ |
| | Follicular hyperplasia | (+) | (+) | (+) | (+) | (+) | — |
| | EMH | (+) | ++ | + | (+) | + | ++ |

Semiquantitative score (according to Gibson-Corley et al [2013]): - none; (+) minimal; + mild; ++ moderate; EMH extramedullary hematopoiesis; PALS periarteriolar lymphoid sheaths.

PCR results for viral DNA in inner organs cannot completely rule out any potential viral spread beyond the nervous system, it is encouraging the further development of cochlear gene therapy based on local administration. We note that we worked with varying amounts of tissue from the different organs. For example, samples from the cochlea for PCR only used surplus cryosections not required for histology. This, together with the subtle AAV spread to the contralateral, non-injected ear, likely explains why we did not detect PCR product there but only in the injected cochlea. This was not a limitation for the inner organs and, hence, even if there was only a minute viral transduction and f-Chrimson expression in the inner organs PCR should have detected the target DNA and our positive control supports the validity of our approach.

From a translational point of view, the AAV administration procedure used here has little potential: in contrast to the largely cartilaginous neonatal mouse cochlea that is readily accessible from a simple retroauricular incision, the human cochlea is largely mature at birth. We chose this widely used approach as it enables high transduction rates (Keppeler et al, 2018; Mager et al, 2018; Bali et al, 2021; review in Dieter et al [2020a]) without the intention of deriving preclinical insights into the biodistribution of the viral particles for regulatory purposes. Rather we consider the approach to mark a worst-case scenario for off-target viral spread after local delivery to the ear. The broad expression of f-Chrimson in both hemispheres with a preference for ipsilateral brainstem areas close to the injected ear are consistent with previous studies such as on the optogenetic manipulation of the retina (Sugano et al, 2016). We consider these findings to indicate the need for preclinical studies on gene therapy of ear and eye to carefully investigate the viral spread within the nervous system. The present study did not detect any obvious histopathology in the brain. Yet, f-Chrimson expression

in the brain could cause a risk of undesired optogenetic neural activation such as stimulation of the facial nerve from an optical cochlear implant, which we have not or rarely observed in our optophysiological studies (Mager et al, 2018; Bali et al, 2021; Huet et al, 2021). Nonetheless, future work en route to clinical optogenetic hearing restoration, should aim to refine the procedure of AAV administration, AAV serotype and promoter to restrict the opsin expression to the SGNs.

In summary, our study demonstrates that expression of f-Chrimson in the auditory nerve endures for the lifetime of a mouse despite the age-related loss of SGNs. There was a spread of the construct within the nervous system without any immunological reactions. Our observation of long-term and relatively stable channelrhodopsin expression with rather mild adverse effects suggests feasibility of late preclinical studies of optogenetic hearing restoration including non-human primates.

# Materials and Methods

### Animals

All animal experiments were carried out in compliance with the relevant national and international guidelines as well as in accordance with German laws governing animal use. The procedures have been approved by the board for animal welfare of the University Medical Center Göttingen and the responsible regional government office (Niedersächsisches Landesamt für Verbraucherschutz und Lebensmittelsicherheit [LAVES]) under the permit number 17/2394.

Wild-type C57BL/6J mice of either sex were used in this study and were euthanized for organ sampling at the ages of 1, 3, 6, 12, and 24 mo, respectively. Animals were kept in a 12-h light/dark cycle with ad libitum access to food and water.

## Viral construct and delivery

C57Bl/6J mice pups (p5-7) were injected unilaterally through the round window membrane using the same batch of viral preparation, i.e., AAV2/6 hSyn-f-Chrimson-eYFP, as described by Mager et al (2018). AAVs were prepared by UNC Virus Core (9.9 × 10^12 vg/ml). A total volume of 1–1.5 $\mu$l was injected using a quartz capillary and pressure of 12.1 PSI in pulses with a duration of 4 ms.

## Organ sampling

Under deep isoflurane anesthesia, mice were decapitated before dissection of brain, cochlea, and internal organs. Whole brains were directly transferred into 4% PFA and left 24 h at +4°C for fixation. After fixation, brains were immersed in 10%, 20%, and 30% cryoprotective sucrose solutions for 24 h at +4°C, respectively. After that, brains were completely embedded in cube molds with OCT compound (Thermo Fisher Scientific) and flash-frozen in liquid nitrogen for storage at −80°C. After brain removal, inner and middle ear structures, that is, cochlea, vestibular system and the ossicular chain, were gently prepared from the respective temporal bones using fine-tip forceps. The protective bony bulla was also removed to visualize the snail shape and round window of the cochleae for sampling. Moreover, tissue samples of a representative organ spectrum including the heart, kidney, liver, lung, and spleen were collected. Each sample was divided with one fraction being frozen at −20°C for PCR analysis and the other fixed in 4% neutral buffered formalin (FA) for histology.

## Histology and immunostaining

### Cochleae
Following the procedure described in Mager et al (2018), cochleae were fixed in 4% FA and decalcified in 0.12 M EDTA for 3–4 d, followed by immersion in 25% glucose and rapid freezing in OCT compound (Thermo Fisher Scientific) for cryoprotection. Serial cryosections of 16 $\mu$m thickness across modiolus were collected and mounted on superfrost glass slides for histology and immunofluorescent staining (IF). A standard hematoxylin and eosin (HE) stain was carried out for histopathological evaluation of bilateral cochlea sections from AAV-injected animals of different age-groups (1, 3, 6, 12, and 24 mo) as well as from PBS-injected (3 mo) and naïve (6 and 12 mo) control animals. For IF, alternating cochlea sections were incubated with guinea pig anti-parvalbumin (1:300; SySy), and chicken anti-GFP mouse (1:500; Abcam) as primary antibodies, and goat anti-guinea pig 568 and goat anti-chicken 488 (1:200; Thermo Fisher Scientific) as secondary antibodies, respectively. Confocal microscopy of immunofluorescence was performed on an SP5 microscope (Leica).

### Brain tissue
Frozen brains from the same AAV-injected (1, 6, 12, and 24 mo) and control animals that were used for histopathologial evaluation of

the cochlea were thawed in 4% neutral buffered formalin and divided into three coronal blocks (I-III) comprising frontal, middle, and hind brain regions, respectively. All three brain blocks were collectively embedded in paraffin and serially sectioned at ~3 $\mu$m with every fifth section being mounted on superfrost slides. Consecutive sections were stained with standard HE for histopathological evaluation, followed by immunohistochemical staining (IHC) with chicken anti-GFP (1:500; Abcam), mouse anti-human CD3 (1:50; Agilent), and mouse anti-IBA1 (1:100; Genetex) primary antibodies, respectively. All IHC procedures were performed using the avidin–biotin peroxidase method on an automated staining system (Ventana Discovery XT) with DAB as the chromogen, and counterstaining with hematoxylin. Brain tissue from a ring-tailed lemur (*Lemur catta*) with cerebral helminthic infection served as a positive control for IHC analysis of microglial activation and density profiling. All HE and IHC slides were scanned with an Aperio CS2 slide scanner (Leica) at maximum magnification of 20× or 40×, and digital images were analyzed with Aperio ImageScope Software (Version 12.3.2.5030; Leica Biosystems).

### Organs
The formalin-fixed lung, heart, liver, kidney, and spleen samples from AAV-injected animals of three different age-groups (6, 12, and 24 mo) as well as from PBS-injected (3 mo) and naïve (6 and 12 mo) control animals were embedded in paraffin, sectioned at ~4 $\mu$m, and stained with HE for histopathological evaluation.

## Immunofluorescence analysis of SGN transduction and density

Somatic immunofluorescence for parvalbumin (PV), being present in most if not all type I SGNs (Shrestha et al, 2018), was used as a neuronal marker, and GFP fluorescence for detection of the marker protein YFP being co-expressed with f-Chrimson. Immunofluorescence was imaged at the confocal level in mid-modiolar cryosections for estimating total (SGN) and f-Chrimson–expressing SGN (GFP+ SGN) counts across the three (apical, middle, basal) cochlear turns (Fig 1A). Using ImageJ, SGN cell bodies were first counted on magenta parvalbumin channel and then they were counted on eYFP one at maximum intensity z-project. Average intensity values from spiral limbus were subtracted from the whole image to reduce background auto-fluorescence. All statistical testing and display of data were carried out on GraphPad Prism. Transduction rate then was determined as:

$$(\textit{Number of double positive SGN somata} \div \textit{Number of parvalbumin positive SGN somata}) \times 100.$$

SGN densities were calculated as:

$$\textit{Number of parvalbumin positive somata} \div \textit{Area of Rosenthal's canal in } 10^4 \ \mu m^2.$$

## Histopathological evaluation

All HE sections were analyzed by a group-blinded, board-certified veterinary pathologist on a BX51 Olympus light microscope with a

Color View I Camera (Olympus). For histological evaluation of the cochleae, five semiquantitative scores (none 0, minimal 1, mild 2, moderate 3, severe 4, see also Table S2) were applied according to the principles described by Gibson-Corley et al (2013) for determining the degree of neuronal density, interstitial vacuolation, and amount of cellular debris in apical, middle and basal turns of spiral ganglions from both ears (injected versus uninjected).

All HE brain sections were likewise scored semiquantitatively for neuronal degeneration and necrosis, inflammatory cells/perivascular cuffing, vacuolation, gliosis, and pigment deposition in both hemispheres of block I (front), II (mid), and III (hind). Correspondingly, HE-stained organ slides were evaluated for tissue-specific histopathological changes, such as inflammatory cell infiltrate, hemorrhage, edema, fibrosis, cellular degeneration and necrosis, hyperplasia/proliferation, and pigment deposition. Every semiquantitative score was individually recorded for each animal in an excel-spreadsheet (Microsoft Office, 2010) with mean values and SD being calculated from all animals per group (raw data are provided in Table S3). Graphical display of data (Figs S1 and S2) was conducted on GraphPad Prism.

### Immunohistochemical quantification of T-cells

In IHC stained serial brain sections, CD3[+] T-cells were separately counted in different annotated cortical and subcortical areas, for example, left hemi-cerebellum, using the pre-set "Counter Tool" function of Aperio ImageScope. Cell numbers were then divided by the annotated area in $mm^2$.

### Positive pixel counts of eYFP-expression and microglia density/activation

To evaluate the degree of f-Chrimson expression and microglia density/activation in digital IHC stained brain sections, the built-in "Positive Pixel Count v9" analysis algorithm of Aperio ImageScope was used, by which the number of positive and negative pixels, as well as annotation areas were extracted. Computational image analysis was applied according to whole slide imaging technology described by Chen et al (2014). This algorithm makes use of the Munsell color scheme where color is defined in three parameters: hue (basic color), chroma (color intensity), and value (notifying lightness/brightness). For brown, the color of chromogen DAB used in this study, hue is given as 0.1 by definition in this algorithm. Chroma, or given as hue width in the program, can be set to 0.33–0.5 which then includes pixels having red and yellow colors. Chroma value was set to 0.33 for microglia activation and 0.2 for f-Chrimson expression samples. Finally, all brown pixels are categorized into three groups according to their brightness, that is, weakly positive brown pixels: 220–175; positive brown pixels: 175–100; strongly positive brown pixels: 100–0. All pixels which are not brown but blue (hematoxylin counterstain) are classified as negative pixels. After obtaining the number of positive and negative pixels, different measures to compare experimental groups and determine microglial

activation were defined. Microglial activation per $μm^2$ was defined as:

$$\frac{Number\ of\ strongly\ positive\ pixels}{Annotated\ area\ in\ μm^2}.$$

### PCR detection

DNA isolation was performed using genomic DNA isolation kit from tissue and cells (Nexttec) according to user's manual. The designed primers targeted an amplicon covering parts of f-Chrimson and eYFP coding sequence. Forward primer: 5'-TTGGGGCATGTACCCAATCC-3'; Reverse primer: 5'-TCTCGTTGGGGTCTTTGCTC-3'. Presence of DNA in the sample was evaluated by PCR for RIM-binding protein-2 using forward primer 5'-GAACATAGGACAGGCAGTACACTTACTTCACCTAG-3' and reverse primer 5'-CAACCTCTCACATACATCACCTGGATCG-3'.

### Statistics

GraphPad Prism 8 was used for all statistical analysis. For comparison of average SGN density between injected and non-injected cochlea, a two-way ANOVA with paired comparison and post-hoc Bonferroni's multiple comparisons test for evaluation of age-specific differences between groups was applied. Age-related average SGN density changes within injected or non-injected cochlea, were evaluated by one-way ANOVAs and post hoc Dunnett's multiple comparisons independently for each group. For comparison of cochlea-turn specific differences in SGN density between injected and non-injected cochlea, two-way ANOVAs with paired comparison and post-hoc Tukey's multiple comparisons for intra-group evaluation of age-related changes were used for each cochlea turn.

# Supplementary Information

# Acknowledgements

We thank Daniela Gerke, Christiane Senger-Freitag, Nadine Dietrich, and Sina Langer for expert technical assistance. This work was funded by the European Research Council through the Advanced Grant "OptoHear" to T Moser under the European Union's Horizon 2020 Research and Innovation program (grant agreement No. 670759) and the Deutsche Forschungsgemeinschaft (DFG, German Research Foundation) under Germany's Excellence Strategy - EXC 2067/1- 390729940 as well as by the Leibniz Program of the DFG (MO896/5) to T Moser. This research was further supported by Fondation Pour l'Audition (FPA RD-2020-10).

## Author Contributions

B Bali: formal analysis, investigation, visualization, methodology, and writing—original draft.

E Gruber-Dujardin: formal analysis, validation, investigation, visualization, methodology, and writing—original draft, review, and editing.

K Kusch: data curation, validation, and writing—review and editing.

V Rankovic: conceptualization, data curation, supervision, investigation, methodology, writing—original draft, and project administration.

T Moser: conceptualization, resources, supervision, funding acquisition, project administration, and writing—original draft, review, and editing.

## Conflict of Interest Statement

T Moser is co-founder of OptoGenTech company.

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
