## [Reviewer comments · Life Science Alliance]

Life Science Alliance

Analyzing efficacy, stability and safety of AAV-mediated optogenetic hearing restoration in mice

Burak Bali, Eva Gruber-Dujardin, Kathrin Kusch, Vladan Rankovic, and Tobias Moser

DOI: <https://doi.org/10.26508/lsa.202101338>

Corresponding author(s): Tobias Moser, University Medical Center Goettingen and Vladan Rankovic, Institute for Auditory Neuroscience and InnerEarLab

Review Timeline:

Submission Date:	2021-12-14
Editorial Decision:	2022-02-04
Revision Received:	2022-03-12
Editorial Decision:	2022-04-08
Revision Received:	2022-04-11
Accepted:	2022-04-12

Scientific Editor: Novella Guidi

Transaction Report:

February 4, 2022

Re: Life Science Alliance manuscript #LSA-2021-01338-T

Prof. Tobias Moser
University Medical Center Goettingen
Institute for Auditory Neuroscience and Collaborative Sensory Research Center 889
InnerEarLab
Robert-Koch-Str. 40
Goettingen, Lower Saxony 37075
Germany

Dear Dr. Moser,

Thank you for submitting your manuscript entitled "Evaluating efficacy, stability and safety of AAV-mediated cochlear channelrhodopsin delivery in mice" to Life Science Alliance. The manuscript was assessed by expert reviewers, whose comments are appended to this letter. As you will note from the reviewers' comments below, all the reviewers are quite supportive and excited about the work that contains important pre-clinical validations. However, they do raise some important concerns that need to be addressed in the manuscript before resubmission. Specifically, for rev#1: the use of B16 mice and how that may confound the analyses needs to be explicitly discussed. Please also make sure to clarify your statistics (rev#1 and rev#2), where concerns like too low number of animals and inappropriate tests conducted are raised by the reviewers. All the other concerns raised by the reviewers should be addressed as well. We, thus, encourage you to submit a revised version of the manuscript back to LSA that responds to all of the reviewers' points.

Thank you for this interesting contribution to Life Science Alliance. We are looking forward to receiving your revised manuscript.

Sincerely,

-- Summary blurb (enter in submission system): A short text summarizing in a single sentence the study (max. 200 characters including spaces). This text is used in conjunction with the titles of papers, hence should be informative and complementary to the title and running title. It should describe the context and significance of the findings for a general readership; it should be

written in the present tense and refer to the work in the third person. Author names should not be mentioned.

B. MANUSCRIPT ORGANIZATION AND FORMATTING:

Reviewer #1 (Comments to the Authors (Required)):

The manuscript from Bali et al. assesses the safety and stability of AAV2/6 mediated channelrhodopsin (ChR) expression after a single round window pressured injection in neonatal mice. The manuscript follows recent work from the same group that validates the use of AAV2/6 to transduce SGNs with F-Chrimson and evoke oABRs in injected animals (Huet et al. 2021). The goal of this work is to provide preclinical evidence that use of optogenetic restoration of sound encoding by optical cochlear implants is safe and reliable.

Bali et al, assess longitudinal expression of F-Chrimson in spiral ganglion, brain and inner organs. Their work demonstrates lifelong expression of F-Chrimson in SGN with higher transduction rate in the apex. While expression is observed in brain regions, no pathology is observed. No viral DNA or pathology was observed in the visceral organs. Those findings from this paper are valuable pre-clinical validation and improving the local expression of AAV-mediated ChR would be an important direction to follow-up in the future studies.

Major questions:

1: choice of animal model. Why did the authors chose to work with a mouse model that suffers from hearing loss (C57B6J), and not a strain like CBA that does not suffer from age related hearing loss?

As stated in the manuscript, C57B6J mice suffer from hearing loss starting ~ 6 months due to Cdh23 mutation which causes hair cell degeneration in the high frequency region and loss of basocochlear SGNs. The reason for using this mouse model might be the relevance with human age-related hearing loss. However, for the purpose of this work, which is to evaluate transduction and survival of basocochlear SGN, the use of C57B6J mice brings up confounding factor contributing to SGN death when evaluating the safety of AAV.

It appears that C57BL6J age-dependent loss of SGN may hide potential toxicity from AAV injection in older animals. The gradient of natural SGN loss is more significant in the high frequency basal region of cochlea. "The age-dependent decline of SGN density seemed to be pronounced in apical and medial modiolar turns of the AAV- injected cochleae from elderly mice of 12 and 24 months." On page 29. This pattern of SGN loss correlates with the gradient of AAV expression. Will that mean AAV is causing some degree of SGN death?

2: Statistical power:

The manuscript is clear and well written, however low number of mice analyzed brings to question the statistical analysis and significance of the work with small "n".

Figure 1B: What is the statistical difference between 1 month and 3 months in the different regions? It seems one mouse had no signal.

Page13: the authors compare SGN density between the injected and contralateral non-injected cochleae using ordinary two-way ANOVA, $p = 0.4374$. What is ordinary 2-way ANOVA? Please clarify whether it means paired ANOVA. The injected cochlea vs. non-injected cochlea is suited for paired t-test for each time point.

"n=2" for PBS and naïve control group are too small compared to AAV injected group (Figure 2B). Thus, any statistical test won't fit into this comparison. Overall, the conclusion of non-significance of SGN loss comparing AAV injection group and naïve group or PBS injection group is improper.

Minor issues:

Introduction:

Page4: Spell out Mio as million

Page5: aimed at hair cell regeneration by transdifferentiation of supporting cells via adenovirus-mediated misexpression of human Atonal transcription factor (Hath1). Is it Atoh1?

Result:

Figure 1A: There is no obvious YFP signal observed in the apical sample at 12 months. Any explanation?

Figure1B: Any reasoning about the higher transduction at apical region compared to the basal region of cochlear? The author argues that the decline of YFP+ SGNs over time is correlated with natural loss of SGN during aging. But the conclusion cannot be drawn from this single piece of data. The author needs to show the percentage of YFP+ SGN among the surviving SGNs, and that ratio should stay almost unchanged over time.

The text mentions on Page11 that the transduction rate is 60% and is shown in the supplementary table2, should be supplementary table1.

Supplementary table1: the description is quite brief and blurry. Please clarify the method, specifically where did N-slices come from? How many samples were included?

Figure2B: please change the text color, e.g., 3 months-virus vs. 3 months PBS looks so closed

Figure3: cell debris is hard to see. Need higher power image and quantification. Include cell death marker or conduct Tunnel staining will provide additional evidence for single cell necrosis/apoptosis.

Reviewer #2 (Comments to the Authors (Required)):

Bali et al. characterize the long term expression and toxicity of AAV-mediated delivery of the excitatory opsin f-Chrimson to targeting spiral ganglion neurons. This work is timely as the number of AAV-base gene therapies are expanding, and more work is needed to better understand the efficacy and limitations of this approach. Overall, the authors do a good job addressing important limitations of this approach. However, additional clarification would improve the manuscript.

Concerns:

1) Important experimental details are missing such as: 1) the volume of viral injection 2) pressure and duration of the pressure pulse used for injection

2) Fig 2

"Mean SGN densities in older age groups (12 and 24 months) tended to be lower in AAV-injected cochleae compared to PBS- or non-injected cochleae (contralateral side or naïve animals) with considerable intra-group variation"

Its not clear from the graph or from this statement whether these differences reached significance

3) Fig 2C

"These differences were obvious but not significant in apical and middle cochlear turns (Figure 2C, ordinary two-way ANOVA, $p = 0.4374$)."

As mentioned the apical and middle regions show a time dependent decline that appears to be "obvious" when compared to the contralateral side, but did not reach significance. Pairwise comparisons between 1 month and 12 & 24 month on the injected side were discussed for average SGN densities, but not for apical and middle regions. Clear differences can also be seen between 1 month and 12 months in the apical (and perhaps middle) section, where virus tends to concentrate. Meanwhile there is no difference between time points in this compartment on the non-injected side. How does one reconcile this? This is likely due to a Type II error (under sampling). The authors should consider performing a post hoc power analysis to see how many samples are necessary to make a meaningful conclusion.

Given this and other data I find their conclusion: "The number of f-Chrimson expressing SGNs declined with age, which primarily reflected an age-dependent loss observed in both experimental and control groups." to be misleading. It is possible and even likely that viral delivery to these areas is accelerating cell loss.

4) Fig 3

There appears to be more hematoxylin stained nuclei in the AAV injected side. Is this observed across other sections. Could these be inflammatory cells?

Perhaps it makes more sense to show histology (H&E and neuronal stains) at 12 months, rather than 24 months, where there seems to be the strongest difference between injected and non-injected sides?

5) Supplemental Fig 1:

Please clarify what the difference is between "neuronal density" as plotted in Supp Fig 1 and PV SGN Density as shown in Fig 2?

6) Conclusion section

"This could imply an additional metabolic demand and/or proteostatic stress for those neurons with continuous opsin expression, which might have contributed to accelerated cellular degeneration over time. Other possible reasons for SGN loss, such as primary neural degeneration, viral toxicity, or neuroimmunological attack of transduced SGNs, seem unlikely since accelerated apical and middle SGN loss only became obvious in old age groups and no histopathological signs of inflammation or increased immune cell infiltration could be observed across all ages."

AAV toxicity can be independent of inflammation (Hirsch et al. 2011 Plos One, Johnston et al. 2021 Elife). It is impossible to know the relative contribution by f-Chrimson versus AAV (or other) toxicity without using AAV expressing YFP alone, AAV expressing no transgene proteins, or empty AAV capsid. Neurotoxic effects of AAV without contributions of the adaptive immune

response have been recently described (Johnston et al 2021 Elife, Suriano et al 2021 bioRxiv).

7) Conclusion section

"Moreover, it seems unlikely that f-Chrimson expression in the brain would cause a risk of undesired broad optogenetic neural activation"

There is little evidence to substantiate this conclusion. To the contrary, multiple studies have shown that even focally delivered light, particularly in the long wavelengths that activate f-Chrimson, spreads throughout the brain and one must restrict expression of opsins to the neural circuits of interest in order to achieve focal activation (Kravitz et al 2013, Brain Research, Danskin et al 2015 PLOS ONE, Allen et al 2015 Learning & Memory).

The authors should discuss this limitation.

Reviewer #3 (Comments to the Authors (Required)):

1) This study provides important data regarding the safety of channelrhodopsins for clinical applications to restore auditory function. The stable long-term expression of the fast-gating Channelrhodopsin (ChR) f-Chrimson delivered by Adeno-associated virus (AAV) in spiral ganglion neurons (SGN) was monitored and statistically analyzed. SGN specific expression of f-Chrimson within the cochlea with a minor decrease in expression over 24 months was detected. The authors detected an age related loss of SGN cells in both control and optogenetic groups with slightly increased levels in f-Chrimson treated cochlea. Therefore, long-term safety of ChR requires further analysis in the future. To avoid the observed systemic spread, the authors suggest optimizing SGN cell specific expression by using different promoters and AAV serotypes. The authors point out that there are still limitations for clinical use, especially regarding the different anatomy of the cochlea in humans and mice at the time of birth. In summary, the comprehensive analysis in the absence of systemic immune responses despite the distribution within the nervous system indicates that f-Chrimson is a potentially safe and promising tool that should be considered for the translation of this biomedical approach.

Overall, the presented long-term data and analysis are important pre-clinical information towards translating this approach to patients. The manuscript is extremely well written, very comprehensive and clear.

2) The introduction is very comprehensive and sound. The Material and Methods are very detailed and there are only a few minor comments (see below).

For all major results, the provided data is strongly supportive.

Synopsis.

What is meant by "ss"AAV cassette? single-stranded DNA?

f-chrimson expression in SGNs upon a single cochlear AAV injection lasts for at least two years.

Supplementary table 1: Why do you present such an excessive number of decimal places?

Supplementary table 2 is mentioned in the text but missing.

Age dependent decline in SGN density is accentuated in apical and middle turns

Figure 2B: no explanation for 3 months PBS sample?

Figure 3 (top right image): What does the circled shape highlight?

f-chrimson expression in the brain following postnatal cochlear AAV-injection

no comments, convincing data and analysis.

f-chrimson expression is not associated with histopathological changes in the brain

excellent data, lots of parameters included.

age-related increase in cerebral T-cell infiltration and activated microglia is independent of cochlear AAV-injection and cerebral f-chrimson expression

excellent data

f chrimson DNA was not detected in inner organs of mice with postnatal cochlear AAV2/6 injection

Figure 7: there is a faint band in negative control. Is this a lane cross-contamination? Also, the primer sets used for lane 3, RIM-binding protein-2 are not listed in the Material and Methods section.

AAV-mediated f-chrimson delivery is not associated with histopathological changes in internal organs

The unbiased analysis includes lots of parameters and is very sound.

Discussion

Virus-mediated optogenetic manipulation of SGNs

good/interesting points raised

The overall context is well discussed.

Spread virus from the cochlea upon pressure injection in the first postnatal week

cautious and comprehensive discussion towards translation, highlighting the different anatomy of human and mouse cochlea at birth

good points raised and good discussion and links to previous work.

Overall, the presented manuscript and data are very sound and there are only minor changes to the text and figures needed, which will improve clarity.

3.

The authors might want to implement a more standard color code/palette for the box-whisker graphs

Please include all animal sample numbers to the figure legends

Rebuttal letter

Reviewer #1 (Comments to the Authors (Required)):

The manuscript from Bali et al. assesses the safety and stability of AAV2/6 mediated channelrhodopsin (ChR) expression after a single round window pressured injection in neonatal mice. The manuscript follows recent work from the same group that validates the use of AAV2/6 to transduce SGNs with F-Chrimson and evoke oABRs in injected animals (Huet et al. 2021). The goal of this work is to provide preclinical evidence that use of optogenetic restoration of sound encoding by optical cochlear implants is safe and reliable. Bali et al, assess longitudinal expression of F-Chrimson in spiral ganglion, brain and inner organs. Their work demonstrates lifelong expression of F-Chrimson in SGN with higher transduction rate in the apex. While expression is observed in brain regions, no pathology is observed. No viral DNA or pathology was observed in the visceral organs. Those findings from this paper are valuable pre-clinical validation and improving the local expression of AAV-mediated ChR would be an important direction to follow-up in the future studies.

First, we would like to thank the reviewer for the appreciation of our work and for the comments that helped us to further improve the MS.

Major questions:

1: choice of animal model. Why did the authors chose to work with a mouse model that suffers from hearing loss (C57B6J),(Mager *et al*, 2018) and not a strain like CBA that does not suffer from age related hearing loss?

As stated in the manuscript, C57B6J mice suffer from hearing loss starting ~ 6 months due to Cdh23 mutation which causes hair cell degeneration in the high frequency region and loss of basocochlear SGNs. The reason for using this mouse model might be the relevance with human age-related hearing loss. However, for the purpose of this work, which is to evaluate transduction and survival of basocochlear SGN, the use of C57B6J mice brings up confounding factor contributing to SGN death when evaluating the safety of AAV.

It appears that C57BL6J age-dependent loss of SGN may hide potential toxicity from AAV injection in older animals. The gradient of natural SGN loss is more significant in the high frequency basal region of cochlea. "The age-dependent decline of SGN density seemed to be pronounced in apical and medial modiolar turns of the AAV- injected cochleae from elderly mice of 12 and 24 months." On page 29. This pattern of SGN loss correlates with the gradient of AAV expression. Will that mean AAV is causing some degree of SGN death?

Thank you very much for this valuable comment. We agree with the reviewer that it will be interesting to perform studies in strains/animal models that are less prone to cochlear degeneration. We chose C57BL6J mice for consistency with mouse work on developing optogenetic hearing restoration out of our lab and the field (e.g. (Duarte *et al*, 2018)). This also applies to preclinical mouse work on gene therapy of the cochlea that uses mice with C57BL6J background. Another reasoning is that efforts for hearing restoration will eventually have to be tested in animal models of human disease, as the scope of the functional outcome, i.e. hearing restoration can only be estimated there. In this regard we have previously used C57BL6J mice as a model of age-related hearing loss and demonstrated that optogenetic stimulation of the auditory nerve remains stable (tested at 9 months of age) while acoustic auditory brainstem responses are progressively reduced (Mager *et al*, 2018). In the same vein, we also consider it valuable to test for stability and safety of AAV-mediated optogenetic manipulation of the SGNs in C57BL6J mice, which might actually

represent an upper bound of treatment associated harm to the cochlea. Yet we largely agree with the reviewer in that disentangling genetic mechanisms of SGN death from harm caused by the AAV manipulation is far from trivial. In fact, we consider this study only a starting point for preclinical assessment of stability and safety, but think that it sets the stage regarding relevant assays and tissue collection for the analysis.

In response to the reviewer's comment, we have done a number of revisions in the MS.

We replaced the sentence on page 29: "Given the greater statistical power of the intra-individual comparison we consider it possible that there might be some adverse effects of AAV-mediated optogenetic manipulation, which is supported by more pronounced histopathological changes of the spiral ganglion in the injected ear." by: "Given the greater statistical power of the intra-individual comparison and with respect to the more pronounced histopathological changes of the spiral ganglion in the injected ear, a potential negative, substance-related long-term effect of the AAV-mediated optogenetic manipulation cannot be ruled out and needs further investigation." and appended: "The age-dependent decline of SGN density seemed to be pronounced in apical and medial turns of the AAV-injected cochleae from elderly mice of 12 and 24 months, which together with the apicobasal gradient could indicate an adverse effect of the AAV-treatment. Future studies in animals lacking basoapically progressing cochlear pathology found in C57BL6J mice will help to evaluate adverse effects of the AAV-treatment." (now on page 31)

We also extended the discussion about this issue on page 31 by adding the following:

"Such effects could imply an additional metabolic demand and/or proteostatic stress for those neurons continuously expressing the opsin and/or the reporter protein YFP, which could contribute to accelerated cellular degeneration and necrosis/apoptosis over time. Other possible reasons for SGN loss would be primary neural degeneration due to the experimental procedure (pressure injection), direct viral toxicity by the injected AAV, as previously described (Hirsch *et al*, 2011; Johnston *et al*, 2021; Suriano *et al*, 2021) or neuroimmunological attack of transduced SGNs. In our study, the observed accelerated SGN loss in apical and middle turns of the ganglion became most evident in the older age groups and was not conspicuous in one and three months old mice. Hence, a primary, injection-related cause or directly AAV-induced apoptosis seem rather unlikely because this might have led to an earlier decline in SGN density. However, for disentangling AAV- or transgene-related effects, future studies could consider additional control groups that received AAVs expressing YFP alone, AAVs expressing no transgenes or empty AAV capsids. With our current analysis, we could not rule out a possible neuroimmunological pathogenesis of accelerated SGN loss in the apical and middle turns as histopathological scoring solely relied on standard HE-stained cochlear sections. However, we consider this rather unlikely since no obvious histopathological signs of inflammation or noticeable leucocyte infiltration could be observed across all age groups. Still, in order to reliably address possible inflammatory cell infiltrates, additional immunostaining of the spiral ganglia with different immune cell markers and their quantification would be necessary, which is planned for our future studies also including non-human primates. "

Furthermore, we further explained our use of C57BL6J mice in the discussion on page 29:

"We chose C57Bl/6J mice as a model for two reasons. First, C57Bl/6J mice are typically employed in mouse mutagenesis. Consequently, most of the available preclinical mouse studies on genetic hearing loss, gene therapy and optogenetic hearing restoration have used this strain such that the results obtained in the present study on C57Bl/6J can be related. Second, C57Bl/6J mice serve as a model of age-related hearing loss due to a mutation in the cadherin-23 gene: are known to start losing hearing by the age of 6 months (Mikaelian *et al*, 1974; Kane *et al*, 2012). Hence, they represent a valuable disease model and our prior work showed that optogenetic stimulation of the auditory nerve was as efficient at the age of 9 months as at 2-3 months, while acoustic hearing was strongly degraded in the older C57Bl/6J mice (Mager *et al*, 2018). This age-related hearing loss

initially manifests itself as degeneration of inner and outer hair cells at the high-frequency sound coding base of cochlea and a subsequent loss, especially of basocochlear SGNs (Someya *et al*, 2009). We consider the use of a model of age-related hearing loss valuable for testing stability and safety of AAV-mediated optogenetic manipulation of the SGNs, but stress that disentangling genetic mechanisms of SGN death from potential harm related to AAV-dosing is far from trivial."

2: Statistical power:

The manuscript is clear and well written, however low number of mice analyzed brings to question the statistical analysis and significance of the work with small "n".

We are aware of this shortcoming. Unfortunately, we could not run another longitudinal series of experiments to increase the statistical basis because Dr. Burak Bali, the key author of the study, left my group and the funding run out. Despite this shortcoming, we consider this study an important starting point for assessing i) the longevity of transgene expression in inner ear cells following viral gene transfer and ii) the biodistribution and safety of gene therapy medicinal products administered to the inner ear. In response to the reviewer's comment we have scrutinized the statistical analysis throughout the MS and removed significance statements where number of observations did not seem to support testing (see below).

Figure 1B: What is the statistical difference between 1 month and 3 months in the different regions? It seems one mouse had no signal.

Yes, indeed, the zero entry resulted from a section where the ganglion of the apical turn had no GFP-positive SGN. We note that these are 16 μm cryosections and this does not mean that there was zero f-Chrimson expressing apical SGN.

Page13: the authors compare SGN density between the injected and contralateral non-injected cochleae using ordinary two-way ANOVA, $p = 0.4374$. What is ordinary 2-way ANOVA? Please clarify whether it means paired ANOVA. The injected cochlea vs. non-injected cochlea is suited for paired t-test for each time point.

In response to the reviewer's comment, we have now clarified the statistical tests applied in a separate methods section and additionally indicated the tests applied with each comparison: "Graph Pad Prism 8 was used for all statistical analysis. For comparison of average SGN density between injected and non-injected cochlea, a two-way ANOVA with paired comparison and post-hoc Bonferroni's multiple comparison test for evaluation of age-specific differences between groups was applied. Age-related average SGN density changes within injected or non-injected cochlea, were evaluated by one-way ANOVAs and post-hoc Dunnett's multiple comparisons independently for each group. For comparison of cochlea-turn specific differences in SGN density between injected and non-injected cochlea, two-way ANOVAs with paired comparison and post-hoc Tukey's multiple comparisons for intra-group evaluation of age-related changes were used for each cochlea turn."

"n=2" for PBS and naïve control group are too small compared to AAV injected group (Figure 2B). Thus, any statistical test won't fit into this comparison. Overall, the conclusion of non-significance of SGN loss comparing AAV injection group and naïve group or PBS injection group is improper.

We have removed the test and now state:

"The average SGN density of 40 SGN bodies/ $10^4 \mu\text{m}^2$ in AAV-injected cochleae from 3 month old mice ($n = 3$) compared to 47.3 SGN bodies/ $10^4 \mu\text{m}^2$ the PBS-injected control ($n = 2$ mice). The average SGN densities of AAV-injected cochleae of 6 - and 12-month-old animals (46.8 and 32.5 SGN

bodies/ $10^4 \mu\text{m}^2$, n = 4 mice and n = 6 mice, respectively) seemed comparable to average SGN densities in age-matched naïve cochleae (43.8 and 39.5 SGN bodies/ $10^4 \mu\text{m}^2$, respectively, n = 2 mice each). “

Minor issues:

Introduction:

Page4: Spell out Mio as million

Done

Page5: aimed at hair cell regeneration by transdifferentiation of supporting cells via adenovirus-mediated misexpression of human Atonal transcription factor (Hath1). Is it Atoh1?

Yes, we changed it to *ATOH1*

Result:

Figure 1A: There is no obvious YFP signal observed in the apical sample at 12 months. Any explanation?

We have aimed for faithfully representing the range of findings. So, yes, this 12 months example only shows few GFP-positive SGNs. We have now added graphical aids to them to serve the easier identification by the reader. For the sake of this rebuttal letter we added 4 further exemplary images of apical sections of the spiral ganglion. (upper left from: Poster_FENSForum2020_Virtual_uploaded-mp3, upper right: and NonclinicalBiosafetyAssessment/Figures/Figure1, lower left and right: this study)

Figure1B: Any reasoning about the higher transduction at apical region compared to the basal region of cochlear? The author argues that the decline of YFP+ SGNs over time is correlated with natural loss of SGN during aging. But the conclusion cannot be drawn from this single piece of data. The author needs to show the percentage of YFP+ SGN among the surviving SGNs, and that ratio should stay almost unchanged over time.

At this point we can only speculate about the tonotopic differences of transduction. We have documented this finding based on larger sample size for the age range of 2-3 months before (e.g. Keppeler et al., EMBO J 2018). Interestingly, AAV-injection into the embryonic otocyst that we employed initially generated a strong basoapical gradient of ChR expression (Hernandez et al., JCI 2014) and postnatal injection into the modiolus resulted in homogenous expression across all tonotopic locations (e.g. Wrobel et al., Sci Translat Med 2018). In response to the reviewer's comment we have now emphasized the possibility of neuron loss due to the optogenetic manipulation throughout the MS: see abstract and discussion.

The text mentions on Page11 that the transduction rate is 60% and is shown in the supplementary table2, should be supplementary table1.
corrected

Supplementary table1: the description is quite brief and blurry. Please clarify the method, specifically where did N-slices come from? How many samples were included?

We clarified, that the same data of cell counting as used for Figure 1 were used and added an additional column in the table to specify the N_{animal} numer and the n_{slice} number:

Age (months)	Injected cochlea				Non-injected cochlea			
	Average	SD	N_{animal}	n_{slices}	Average	SD	N_{animal}	n_{slices}
1	58.9	26.8	4	12	3.48	5.26	4	12
3	32.5	36.2	3	9	2.01	3.72	3	9
6	46.6	31.3	3	9	1.43	1.90	3	8
12	42.4	31.1	6	18	3.30	5.54	6	18
24	34.1	31.3	3	9	8.05	14.2	3	9

Supplementary Table 1: Average share of transduced SGN in percent across cryosections

The share of SGNs expressing f-Chrimson-eYFP expressed as percentage of all parvalbumin-positive SGNs for the injected and non-injected cochlear of the 5 age-groups investigated here. Same data of SGN cell counts as in Figure 1. In general, 3 slices per cochlea were counted (one exception due to technical reasons).

Figure2B: please change the text color, e.g., 3 months-virus vs. 3 months PBS looks so closed
Done

Figure3: cell debris is hard to see. Need higher power image and quantification. Include cell death marker or conduct TUNEL staining will provide additional evidence for single cell necrosis/apoptosis.

In response to the comment we replaced the images of figure 3 by those of another 12-months old mouse, where the neural loss was drastic and so morphological changes in the injected spiral ganglion are more obvious. Unfortunately, no additional staining (e.g. TUNEL assay or Caspase 3 immunohistochemistry) for the detection of dead neurons could be applied due to a limited number of representative cochlear cryosections available per animal, but this will be considered for future experiments.

Figure 3 Histopathological analysis of the cochlea.

Light micrographs of a HE stained cochleae of a 12-month-old mouse (#652279): AAV-injected (A) side showing an extreme example of SGN loss (circle represents one of the remaining SGN), accompanied by interstitial vacuolation (asterisk) and some cellular debris (arrows) in the apical (a) and middle (m) modiolar turns, compared to the contralateral, non-injected side (B). Histological appearance of basal (b) parts was more similar on both sides; scale bars left: 200 μm; right: 50 μm.

For quantification of cellular debris, the same ordinal semi quantitative scores (none 0, minimal 1, mild 2, moderate 3, severe 4) were applied that were also used for the other histomorphological parameters investigated (neuronal density, interstitial vacuolation) according to the principles described by (Gibson-Corley et al, 2013). This analysis is summarized in Supplementary Figure 1. For further detail, we added the following supplementary table 2 to the manuscript:

Score	Spiral ganglion (% tissue affected)
0	None
1	<25
2	26-50
3	51-75
4	76-100

Supplementary table 2 Semi quantitative, ordinal scores based on the distribution of the spiral ganglion findings (according to Gibson-Corley et al., 2013)

Reviewer #2 (Comments to the Authors (Required)):

Bali et al. characterize the long term expression and toxicity of AAV-mediated delivery of the excitatory opsin f-Chrimson to targeting spiral ganglion neurons. This work is timely as the number of AAV-base gene therapies are expanding, and more work is needed to better understand the efficacy and limitations of this approach. Overall, the authors do a good job addressing important limitations of this approach. However, additional clarification would improve the manuscript.

First, we would like to thank the reviewer for the appreciation of our work and for the comments that helped us to further improve the MS. 1–1.5 II

Concerns:

1) Important experimental details are missing such as: 1) the volume of viral injection 2) pressure and duration of the pressure pulse used for injection

Done: details added to MS: volume: 1–1.5 μ l, pressure of 12.1 PSI in pulses with a duration of 4 ms.

2) Fig 2

"Mean SGN densities in older age groups (12 and 24 months) tended to be lower in AAV-injected cochleae compared to PBS- or non-injected cochleae (contralateral side or naïve animals) with considerable intra-group variation"

Its not clear from the graph or from this statement whether these differences reached significance

As we stated in the original submittal this did not reach statistical significance. In response to the reviewers comment we have revised the statement for further clarity, page 14:

"Mean SGN densities in older age groups (12 and 24 months) tended to be lower for apical and middle cochlear turns of AAV-injected cochleae compared to non-injected cochleae. Considerable intra-group variation together with the small sample size might have kept us from documenting statistical significance for the apical turn, but was detected for the middle cochlea turn (Figure 2C, two-way ANOVA with paired comparison for apical turn: $p = 0.0883$ and for middle $p = 0.0336$). "

3) Fig 2C

"These differences were obvious but not significant in apical and middle cochlear turns (Figure 2C, ordinary two-way ANOVA, $p = 0.4374$)."

As mentioned the apical and middle regions show a time dependent decline that appears to be "obvious" when compared to the contralateral side, but did not reach significance. Pairwise comparisons between 1 month and 12 & 24 month on the injected side were discussed for average SGN densities, but not for apical and middle regions. Clear differences can also be seen between 1 month and 12 months in the apical (and perhaps middle) section, where virus tends to concentrate. Meanwhile there is no difference between time points in this compartment on the non-injected side. How does one reconcile this? This is likely due to a Type II error (under sampling). The authors should consider performing a post hoc power analysis to see how many samples are necessary to make a meaningful conclusion.

Given this and other data I find their conclusion: "The number of f-Chrimson expressing SGNs declined with age, which primarily reflected an age-dependent loss observed in both experimental and control groups." to be misleading. It is possible and even likely that viral delivery to these areas is accelerating cell loss.

We are aware of the shortcoming of small sample size. Unfortunately, we could not run another longitudinal series of experiments to increase the statistical basis because Dr. Burak Bali, the key author of the study, left my group and the funding ran out. In response to the reviewer's comment we have scrutinized the statistical analysis throughout the MS and removed significance statements where number of observations did not seem to support testing. The latter, for example, applied for comparison to PBS-injected and naïve animals.

"The average SGN density of 40 SGN bodies/ $10^4 \mu\text{m}^2$ in AAV-injected cochleae from 3 month old mice ($n = 3$) compared to 47.3 SGN bodies/ $10^4 \mu\text{m}^2$ the PBS-injected control ($n = 2$ mice). The average SGN densities of AAV-injected cochleae of 6 - and 12-month-old animals (46.8 and 32.5 SGN bodies/ $10^4 \mu\text{m}^2$, $n = 4$ mice and $n = 6$ mice, respectively) seemed comparable to average SGN densities in age-matched naïve cochleae (43.8 and 39.5 SGN bodies/ $10^4 \mu\text{m}^2$, respectively, $n = 2$ mice each)."

4) Fig 3

There appears to be more hematoxylin stained nuclei in the AAV injected side. Is this observed across other sections. Could these be inflammatory cells?

We agree with the reviewer about this consideration and had paid an effort to further immunohistochemistry in this direction. Unfortunately, our immunostaining with several inflammatory cell markers (IBA-1, CD3, Pax5) did not work reliably and we only had a limited number of cochlear sections available per animal. This is why, we finally decided to present the morphological only findings from the H&E results. There, with semi-quantitative estimation it seems that the impression of hypercellularity reflects the reduction of the larger spiral ganglion neurons rather than a real increase in glial and/or inflammatory cells. But, in order to include and point out this very important aspect in our manuscript, we appended the following statement to the discussion section:

"With our current analysis, we could not rule out a possible neuroimmunological pathogenesis of accelerated SGN loss in the apical and middle turns as histopathological scoring solely relied on standard HE-stained cochlear sections. However, we consider this rather unlikely since no obvious histopathological signs of inflammation or noticeable leucocyte infiltration could be observed across all age groups. Still, in order to reliably address possible inflammatory cell infiltrates, additional immunostaining of the spiral ganglia with different immune cell markers and their quantification would be necessary, which is planned for our future for our future studies also including non-human primates."

Perhaps it makes more sense to show histology (H&E and neuronal stains) at 12 months, rather than 24 months, where there seems to be the strongest difference between injected and non-injected sides?

In response to the reviewer's comment we replaced the images of figure 3 by those of another 12-months old mouse, where the neural loss was drastic and so morphological changes in the injected spiral ganglion are more obvious.

We also provide a panel with micrographs of HE stained cochleae (injected vs. non-injected side) from all mice analyzed to the supplementary material (Suppl. Table 3)

Figure 3 Histopathological analysis of the cochlea.

Light micrographs of a HE stained cochleae of a 12-month-old mouse (#652279): AAV-injected (A) side showing an extreme example of SGN loss (circle represents one of the remaining SGN), accompanied by interstitial vacuolation (asterisk) and some cellular debris (arrows) in the apical (a) and middle (m) modiolar turns, compared to the contralateral, non-injected side (B). Histological appearance of basal (b) parts was more similar on both sides; scale bars left: 200 μm ; right: 50 μm .

5) Supplemental Fig 1:

Please clarify what the difference is between "neuronal density" as plotted in Supp Fig 1 and PV SGN Density as shown in Fig 2?

"Neuronal density" (Supp Fig 1) is one of the histomorphological parameters (besides interstitial vacuolation and cellular debris) that were analyzed by semi quantitative scoring (0, 1, 2, 3, 4) within HE stained sections according to the principles described by (Gibson-Corley et al, 2013). It is the estimated percentage of neurons within spiral ganglion tissue while 100% resembles normal neuronal density in unaffected control sections. For further clarification, we added the following supplementary table 2 to the manuscript:

Score	Spiral ganglion (% tissue affected)
0	None
1	<25
2	26-50
3	51-75
4	76-100

Supplementary table 2 Semi quantitative, ordinal scores based on the distribution of the spiral ganglion findings (according to Gibson-Corley et al., 2013)

PV SGN density (i.e. density of parvalbumin-positive spiral ganglion neurons), shown in Fig. 2, was calculated by the counted number of cell somata that showed a positive immunofluorescent signal for the neuronal marker parvalbumin (PV+) divided by the Area of Rosenthal's canal in $10^4 \mu\text{m}^2$ (see also Methods section, page 8). In response to the reviewer's comment, we now detail this also in the legend to Fig. 2.

6) Conclusion section

"This could imply an additional metabolic demand and/or proteostatic stress for those neurons with continuous opsin expression, which might have contributed to accelerated cellular degeneration over time. Other possible reasons for SGN loss, such as primary neural degeneration, viral toxicity, or neuroimmunological attack of transduced SGNs, seem unlikely since accelerated apical and middle SGN loss only became obvious in old age groups and no histopathological signs of inflammation or increased immune cell infiltration could be observed across all ages."

AAV toxicity can be independent of inflammation (Hirsch et al. 2011 Plos One, Johnston et al. 2021 Elife). It is impossible to know the relative contribution by f-Chrimson versus AAV (or other) toxicity without using AAV expressing YFP alone, AAV expressing no transgene proteins, or empty AAV capsid. Neurotoxic effects of AAV without contributions of the adaptive immune response have been recently described (Johnston et al 2021 Elife, Suriano et al 2021 bioRxiv).

In response to the reviewer's comment, we have quoted these important papers and we revised discussion on page 30:

"Given the greater statistical power of the intra-individual comparison and with respect to the more pronounced histopathological changes of the spiral ganglion in the injected ear, a potential negative, substance-related long-term effect of the AAV-mediated optogenetic manipulation cannot be ruled out and needs further investigation. The age-dependent decline of SGN density seemed to be pronounced in apical and medial turns of the AAV-injected cochleae from elderly mice of 12 and 24 months, which together with the apicobasal gradient could indicate an adverse effect of the AAV-treatment. Future studies in animals lacking basoapically progressing cochlear pathology found in C57BL6J mice will help to further evaluate adverse effects of the AAV-treatment. Such effects could imply an additional metabolic demand and/or proteostatic stress for those neurons continuously expressing the opsin and/or the reporter protein YFP, which could contribute to accelerated cellular degeneration and necrosis/apoptosis over time. Other possible reasons for SGN loss would be primary neural degeneration due to the experimental procedure (pressure injection), direct viral toxicity by the injected AAV, as previously described (Hirsch *et al*, 2011; Johnston *et al*, 2021; Suriano *et al*, 2021) or neuroimmunological attack of transduced SGNs."

7) Conclusion section

"Moreover, it seems unlikely that f-Chrimson expression in the brain would cause a risk of undesired broad optogenetic neural activation"

There is little evidence to substantiate this conclusion. To the contrary, multiple studies have shown that even focally delivered light, particularly in the long wavelengths that activate f-Chrimson, spreads throughout the brain and one must restrict expression of

opsins to the neural circuits of interest in order to achieve focal activation (Kravitz et al 2013, Brain Research, Danskin et al 2015 PLOS ONE, Allen et al 2015 Learning & Memory). The authors should discuss this limitation.

Done

“The broad expression of f-Chrimson in both hemispheres with a preference for ipsilateral brainstem areas close to the injected ear are consistent with previous studies such as on the optogenetic manipulation of the retina (Sugano *et al*, 2016). We consider these findings to indicate the need for preclinical studies on gene therapy of ear and eye to carefully investigate the viral spread within the nervous system. The present study did not detect any obvious histopathology in the brain. Yet, f-Chrimson expression in the brain could cause a risk of undesired optogenetic neural activation such as stimulation of the facial nerve from an optical cochlear implant, which we have not or rarely observed in our optophysiological studies (Mager *et al*, 2018; Bali *et al*, 2021; Huet *et al*, 2021). Nonetheless, future work *en route* to clinical optogenetic hearing restoration, should aim to refine the procedure of AAV administration, AAV serotype and promoter to restrict the opsin expression to the SGNs.”

Reviewer #3 (Comments to the Authors (Required)):

First, we would like to thank the reviewer for the appreciation of our work and for the comments that helped us to further improve the MS.

1) This study provides important data regarding the safety of channelrhodopsins for clinical applications to restore auditory function. The stable long-term expression of the fast-gating Channelrhodopsin (ChR) f-Chrimson delivered by Adeno-associated virus (AAV) in spiral ganglion neurons (SGN) was monitored and statistically analyzed. SGN specific expression of f-Chrimson within the cochlea with a minor decrease in expression over 24 months was detected. The authors detected an age related loss of SGN cells in both control and optogenetic groups with slightly increased levels in f-Chrimson treated cochlea. Therefore, long-term safety of ChR requires further analysis in the future. To avoid the observed systemic spread, the authors suggest optimizing SGN cell specific expression by using different promoters and AAV serotypes. The authors point out that there are still limitations for clinical use, especially regarding the different anatomy of the cochlea in humans and mice at the time of birth. In summary, the comprehensive analysis in the absence of systemic immune responses despite the distribution within the nervous system indicates that f-Chrimson is a potentially safe and promising tool that should be considered for the translation of this biomedical approach.

Overall, the presented long-term data and analysis are important pre-clinical information towards translating this approach to patients. The manuscript is extremely well written, very comprehensive and clear.

2) The introduction is very comprehensive and sound. The Material and Methods are very detailed and there are only a few minor comments (see below).
For all major results, the provided data is strongly supportive.

Synopsis.

What is meant by "ss" AAV cassette? single-stranded DNA?

Done, yes, single-stranded DNA was meant but removed for simplicity from synopsis

f-chrimson expression in SGNs upon a single cochlear AAV injection lasts for at least two years.

Supplementary table 1: Why do you present such an excessive number of decimal places?

Done, rounded to 3 numbers of digit

Supplementary table 2 is mentioned in the text but missing.

added

Age dependent decline in SGN density is accentuated in apical and middle turns

Figure 2B: no explanation for 3 months PBS sample?

added

Figure 3 (top right image): What does the circled shape highlight?

The circle highlights one of the remaining SG neurons. We forgot to mention that in the figure legend and inserted the corresponding explanation. (Please note that we also have changed the entire Figure 3 by micrographs from a 12-months-old mouse, where the morphological changes in the injected spiral ganglion are more pronounced).

f-chrimson expression in the brain following postnatal cochlear AAV-injection
no comments, convincing data and analysis.

f-chrimson expression is not associated with histopathological changes in the brain
excellent data, lots of parameters included.

age-related increase in cerebral T-cell infiltration and activated microglia is independent of cochlear AAV-injection and cerebral f-chrimson expression
excellent data

f chrimson DNA was not detected in inner organs of mice with postnatal cochlear AAV2/6 injection

Figure 7: there is a faint band in negative control. Is this a lane cross-contamination? Also, the primer sets used for lane 3, RIM-binding protein-2 are not listed in the Material and Methods section.

added

AAV-mediated f-chrimson delivery is not associated with histopathological changes in internal organs

The unbiased analysis includes lots of parameters and is very sound.

Discussion

Virus-mediated optogenetic manipulation of SGNs

good/interesting points raised

The overall context is well discussed.

Spread virus from the cochlea upon pressure injection in the first postnatal week

cautious and comprehensive discussion towards translation, highlighting the different anatomy of human and mouse cochlea at birth

good points raised and good discussion and links to previous work.

Overall, the presented manuscript and data are very sound and there are only minor changes to the text and figures needed, which will improve clarity.

3.

The authors might want to implement a more standard color code/palette for the box-whisker graphs

In response to the reviewer's suggestion, we tried to identify an alternative color scheme to use for graphs in this manuscript. We offer to change the graphs as indicated exemplarily for figure 2B below, but have some concerns regarding color-blindness visibility of the new graphs and would be glad for recommendations.

Current color scheme:

Potential new color scheme:

Please include all animal sample numbers to the figure legends
added

April 8, 2022

RE: Life Science Alliance Manuscript #LSA-2021-01338-TR

Prof. Tobias Moser
University Medical Center Goettingen
Institute for Auditory Neuroscience and Collaborative Sensory Research Center 889
InnerEarLab
Robert-Koch-Str. 40
Goettingen, Lower Saxony 37075
Germany

Dear Dr. Moser,

Thank you for submitting your revised manuscript entitled "Analyzing efficacy, stability and safety of AAV-mediated optogenetic hearing restoration in mice". We would be happy to publish your paper in Life Science Alliance pending final revisions necessary to meet our formatting guidelines.

- Please upload all figure files as individual ones, including the supplementary figure files; all figure legends should only appear in the main manuscript file
- please upload the Graphical Abstract as a single file as well, with the file designation: Graphical Abstract
- please upload your Tables in editable .doc or excel format
- main tables can be included at the bottom of the main manuscript file or be sent as separate files.
- Supporting Tables should be uploaded separately
- please consult our manuscript preparation guidelines <https://www.life-science-alliance.org/manuscript-prep> and make sure your manuscript sections are in the correct order
- please add your main, supplementary figure, and table legends to the main manuscript text after the references section
- please be sure that all authors are added to the Authors' contribution section in the manuscript text
- please add callouts for Figures 3A-B, 6B, and 7A-B to your main manuscript text
- scale bars are hardly visible in Figure 4A, please mark them in black

A. FINAL FILES:

B. MANUSCRIPT ORGANIZATION AND FORMATTING:

Sincerely,

Reviewer #1 (Comments to the Authors (Required)):

The authors were aware of the limitations and did a good job revising the manuscript to deliver a more faithful explanation. Please keep in mind the small sample size may weaken the overall conclusion.

Reviewer #2 (Comments to the Authors (Required)):

While the manuscript still contains a number of shortcomings primarily due to the limited number of mice and limited number of tissue sections to complete a truly thorough analysis, the main observations are useful and important to disseminate to the community of researchers working on AAV and gene therapies. I believe that the authors addressed the majority of the shortcomings adequately given the limitations they described.

April 12, 2022

RE: Life Science Alliance Manuscript #LSA-2021-01338-TRR

Prof. Tobias Moser
University Medical Center Goettingen
Institute for Auditory Neuroscience and Collaborative Sensory Research Center 889
InnerEarLab
Robert-Koch-Str. 40
Goettingen, Lower Saxony 37075
Germany

Dear Dr. Moser,

Thank you for submitting your Research Article entitled "Analyzing efficacy, stability and safety of AAV-mediated optogenetic hearing restoration in mice". It is a pleasure to let you know that your manuscript is now accepted for publication in Life Science Alliance. Congratulations on this interesting work.

DISTRIBUTION OF MATERIALS:

Again, congratulations on a very nice paper. I hope you found the review process to be constructive and are pleased with how the manuscript was handled editorially. We look forward to future exciting submissions from your lab.

Sincerely,
